# Late-stage guanine C8−H alkylation of nucleosides, nucleotides, and oligonucleotides via photo-mediated Minisci reaction

Ruoqian Xie[1,2,4], Wanlu Li[1,4], Yuhua Ge[1] ✉, Yutong Zhou[2,3], Guolan Xiao[2], Qin Zhao[2], Yunxi Han[2], Yangyan Li[2] & Gang Chen [2,3] ✉

Chemically modified nucleosi(ti)des and functional oligonucleotides (ONs, including therapeutic oligonucleotides, aptamer, nuclease, etc.) have been identified playing an essential role in the areas of medicinal chemistry, chemical biology, biotechnology, and nanotechnology. Introduction of functional groups into the nucleobases of ONs mostly relies on the laborious de novo chemical synthesis. Due to the importance of nucleosides modification and aforementioned limitations of functionalizing ONs, herein, we describe a highly efficient site-selective alkylation at the C8-position of guanines in guanosine (together with its analogues), GMP, GDP, and GTP, as well as late-stage functionalization of dinucleotides and single-strand ONs (including ssDNA and RNA) through photo-mediated Minisci reaction. Addition of catechol to assist the formation of alkyl radicals via in situ generated boronic acid catechol ester derivatives (BACED) markedly enhances the yields especially for the reaction of less stable primary alkyl radicals, and is the key to success for the post-synthetic alkylation of ONs. This method features excellent chemoselectivity, no necessity for pre-protection, wide range of substrate scope, various free radical precursors, and little strand lesion. Downstream applications in disease treatment and diagnosis, or as biochemical probes to study biological processes after linking with suitable fluorescent compounds are expected.

Nucleoside and nucleotide analogs, mimicking the natural nucleosides, represent a large family of anti-tumor and anti-virus drugs, such as FDA-approved anti-Covid-19 drug Remdesivir and Molnupiravir[1–3]. Nucleotides, such as 5′-triphosphates, are fundamental building blocks for the polymerase-mediated synthesis of nucleic acid[4,5]. Moreover, base-functionalized oligonucleotides are of great interest due to their applications in the field of drug discovery and chemical biology[6,7]. Given the importance of these biomolecules, various methods have been developed to access these compounds and their analogs[8–10]. In general, multiple synthetic steps are required for preparing these biologically active nucleosi(ti)de analogs[8], including de novo synthesis via glycosylation of ribose with base, and multiple steps of chemical

[1]School of Chemistry and Chemical Engineering, Southeast University, Nanjing 211189, People's Republic of China. [2]Shanghai Key Laboratory for Molecular Engineering of Chiral Drugs, School of Chemistry and Chemical Engineering, Shanghai Jiao Tong University, Shanghai 200240, People's Republic of China. [3]Key Laboratory of Green and High-End Utilization of Salt Lake Resources, Chinese Academy of Sciences, Qinghai Institute of Salt Lakes, Chinese Academy of Sciences, Xining 810008 Qinghai, People's Republic of China. [4]These authors contributed equally: Ruoqian Xie, Wanlu Li. ✉e-mail: geyuhua@seu.edu.cn; gchen2018@sjtu.edu.cn

modification from protected nucleosides. Especially for the base-functionalized oligonucleotides synthesis[9,10], solid-phase DNA synthesis based on phosphoramidite chemistry or DNA polymerases method with 5′-triphosphorylated nucleosides is always needed, which may have a compatibility issue with unnatural nucleoside monomer. Therefore, the development of efficient, mild, and water-compatible methods for the direct or late-stage functionalization of nucleosi(ti) des and oligonucleotides is highly desirable[11,12], which strategy has been widely used in the modification of drug molecules, natural products, and biological macromolecules[13–15].

C8 functionalized purines possess unique biological activities. For example, C8-arylated, alkenylated, and alkynlated analogs are important fluorescent nucleobases as conjugated scaffolds of purine structures[7]. Moreover, the C8-alkylated analogs exhibit antiviral or anticancer activities (Fig. 1a)[16–18]. To be more specific, 8-methyladenosine is a potent and highly selective inhibitor of the vaccinia virus, and 8-ethyl adenosine shows special inhibition against respiratory syncytial virus (RSV)[16]. C8-substituted guanosine triphosphate (GTP) analogs efficiently inhibit the polymerization of FtsZ and could be a promising class of novel antibiotics[17], while 8-cyclopentyl-2,6-diphenyl purine is an adenosine receptor antagonist[18]. Hence, great efforts have been made to the synthesis of C8-alkylated purine nucleosides. As an early example, the classical transition-metal catalyzed cross-coupling strategy was presented to construct the C8-alkylpurine frames from 8-halogenopurine nucleosides (see Supplementary Fig. 7)[19]. Notably, Hocek group reported the direct methylation of deoxyadenosine triphosphate (dATP) via Suzuki coupling, providing the C8-Me dATP[20]. Moreover, the radical process, as a prevailing method for alkylation reaction, always proceeds under mild reaction conditions. However, there are few examples of radical alkylation of C8-H purines that have been reported to date (Fig. 1b)[21–24]. In the 1970s, Kawazoe and Pless group reported direct modification of guanosine using a classic radical approach, however, the yield is low (<30%), and substrate scope is also limited (see Supplementary Fig. 7)[21,22]. In 2012, Qu and Guo group successfully accomplished direct C8 radical alkylation of protected purines with cycloalkane promoted by tBuOOtBu delivering an 8-cycloalkyl purine skeleton at 140 °C[23]. Later, Zard and co-workers described an effective xanthate-based alkylation of the Ac group-protected guanosine derivative[24]. Meanwhile, very limited examples of selective functionalization at purine in DNA/RNA were reported[25–30] (Fig. 1c). One approach is based on pre-functionalized oligonucleotides. In 2010, Manderville group reported the efficient C8 arylation of DNA oligonucleotides via Suzuki-Miyaura cross-coupling of C8-Br G[25]. In 2022, Zhou group achieved the $S_NAr$ reaction of C6-I purine using different nucleophile regents[26]. Another approach is late-stage functionalization of nucleic acid with non-canonical nucleosides. In 2019, Wang and Cheng group[27] reported photochemical demethylation of N6-Methyladenosine (N6-mA) residues in RNA by Flavin mononucleotide (FMN), which is important in epigenetic research. Later, Balasubramanian and Gaunt group reported a selective chemical functionalization of N6-mdA in DNA[28] via visible light-mediated photoredox catalysis. Direct modification of natural nucleic acid is more challenging due to the similar functional group in the nucleobases. In 2020, Saraogi group achieved selective functionalization at the N2-position of guanosine in oligonucleotide via reductive amination owing to the higher reactivity of N2 amine of guanosine than other nucleobases (adenine and cytosine)[29]. In the same year, He and co-workers reported fast and reversible labeling of single-stranded guanine bases in live cells using a new $N_3$-kethoxal reagent for RNA secondary structure mapping[30].

Despite the above-mentioned efforts, deficiencies in the alkylation procedures have yet to be overcome, which include harsh conditions, lengthy steps, limited substrate scope, and poor product diversity. In particular, fewer methods are available for the selective functionalization at the C8 site of G in RNA/DNA oligonucleotides. In connection with our continuous interest in the late-stage functionalization of drug molecules, natural products, and biomacromolecules[31,32], we wish to report our investigations toward the direct alkylation at the C8 site of nucleosi(ti)de analogs via catechol-promoted photo-mediated Minisci reaction with alkylboronic acids and their derivatives (Fig. 1d). Moreover, late-stage alkylation at the C8-site of G in RNA/DNA oligonucleotides has also been achieved, which would have the potential to be further applied in drug development and biological studies.

## Results
### Reaction development
We commenced our research by investigating the C8-alkylation of unprotected guanosine **1a** under several reported reaction conditions (see Supplementary Fig. 17)[23,24,33]. However, the desired C8-Et guanosine **3a** was obtained in 23% yield only under the condition developed by Monlander group with EtBF$_3$K **2a** as radical precursors: [MesAcr (5 mol%), (NH$_4$)$_2$S$_2$O$_8$ (2.0 equiv.), and TFA (1.0 equiv.) in MeCN/H$_2$O (1/1) under white light (85 W) irradiation at room temperature] (Table 1, **entry 1**)[33]. These results further demonstrated the challenge of late-stage functionalization of unprotected guanosine. Then, extensive conditions were screened to optimize the reaction. Neither changing the photocatalyst nor solvent gave detectable improvement. (see Supplementary Tables 3 and 4 full optimization). When the loading of alkylating reagent increased from 2 equiv. to 4 equiv., the yield increased to 37% (Table 1, **entry 2**). However, no better results were obtained after other condition screenings (see Supplementary Table 5 for full optimization). The oxidative potential of RBF$_3$K is lower than the corresponding RB(OH)$_2$, and RBPin, Gouverner, and Davis group reported that the addition of catechol would in situ-generate boronic acid catechol ester derivative (BACED) reagents with reduced oxidative potential to promote the formation of alkyl radicals[34–36]. Indeed, a much higher yield was obtained when catechol was used as an additive in our reaction (Table 1, **entry 3**). Moreover, ethylboronic acid (EtB(OH)$_2$, **2a**) acting as the radical precursor, which is much cheaper than EtBF$_3$K and not compatible with the reported condition[33], proceeded well with a similar yield (87%, Table 1, **entry 4**). Whereas, by using other additives such as phenol, hydroquinone, and ethylene glycol, the desired alkylated product was in much lower yields (Table 1, **entries 5-7**). Other analogs with an electron-donating group of catechol present the same effect (see Supplementary Table 7 for full optimization). Further control experiments proved that catechol and oxidant are essential for this transformation (Table 1, **entries 8, 9**). This reaction can also deliver the desired product in 74% yield in the absence of trifluoroacetic acid (TFA), which could be further applied to the acid-sensitive substrates (Table 1, **entry 10**). Because the RB(OH)$_2$ or RBF$_3$K with special functional groups are not commercially available, and need to be synthesized from the corresponding RBpin, so RBpin as radical precursors were tested to expand the range of substrate adaptability, which may avoid the additional functional group transformations. Adding methylboronic acid (MeB(OH)$_2$), in situ generating the EtB(OH)$_2$ from the EtBPin, delivered the desired product **3a** in high yield (**Table 1, entry11**)[37]. This is facilitated by the fact that our protocol, with MeB(OH)$_2$ as the substrate, is essentially free of conversion (see Supplementary Table 8). Further control experiments proved that catechol and MeB(OH)$_2$ are essential for this transformation (Table 1, **entries 12, 13**).

In the presence of radical scavenger 2,2,6,6-tetramethylpiperidinooxy (TEMPO), the alkylation reaction was completely inhibited and the radical-trapping byproduct **2j-1** was isolated in 44% yield (see Supplementary Fig. 21). Based on observed results and the

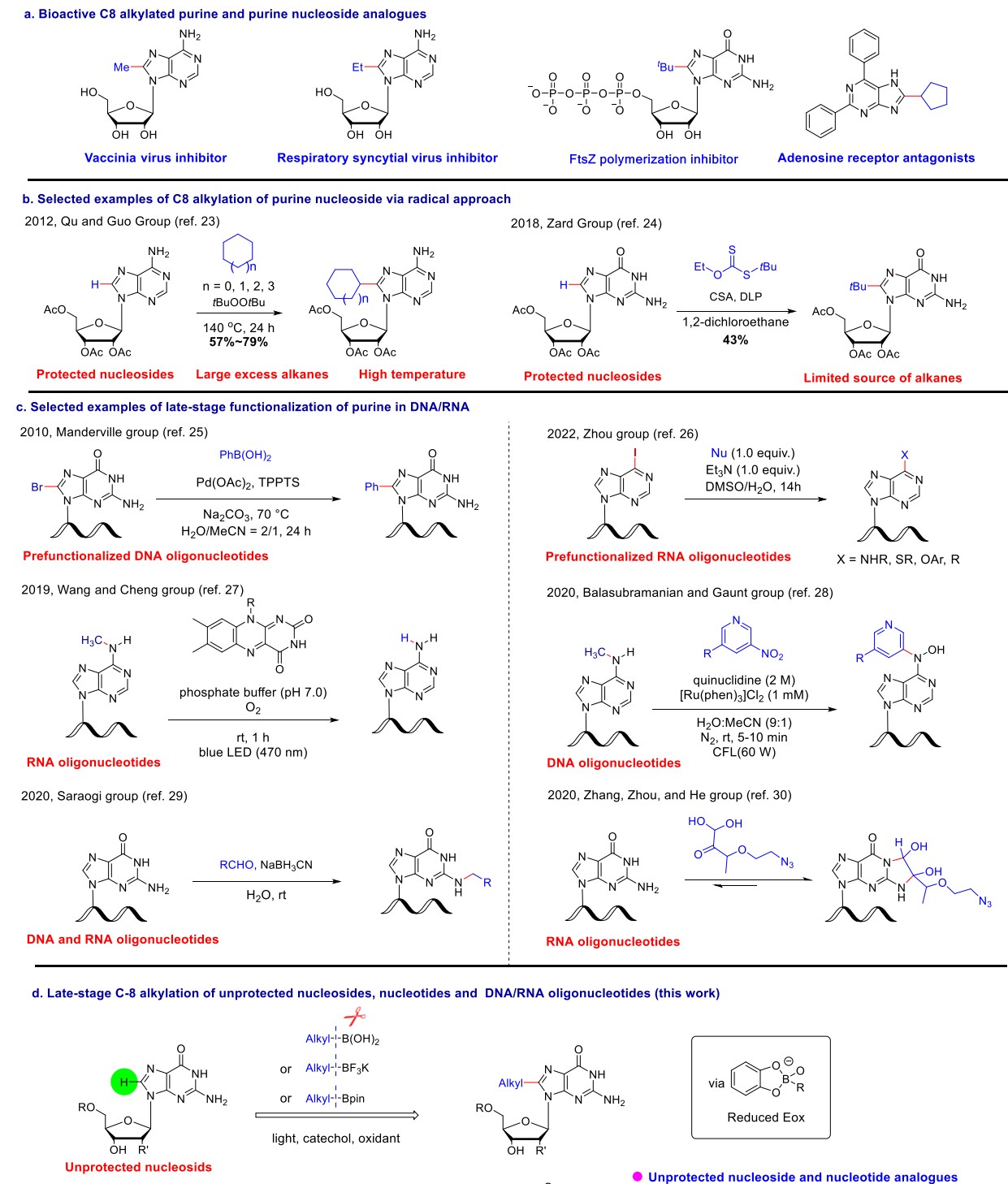

**Fig. 1 | Background for reaction development. a** Representative examples of bioactive C-8 alkylated nucleosides. **b** Selected examples of C8 alkylation of purine. **c** Selected examples of late-stage functionalization of purine of DNA/RNA. **d** Our

strategy for late-stage functionalization of C-8 alkylated nucleosides, nucleotides, and RNA & DNA oligonucleotides. Me methyl, Et ethyl, *t*Bu *tert*-butyl, Ac Acetyl, HMDS hexamethyldisilazane, CSA camphorsulfonic acid, DLP dilauroyl peroxide.

previous reports[34], a possible radical mechanism was proposed (see Supplementary Fig. 22). Initially, alkylboronic acids and their derivative reacted with catechol to deliver the boronic acid catechol ester derivative (BACED) with reduced oxidative potential. This in situ formed

BACED reagent advantageously generated RCH2• radical via Mesacr-catalyzed SET way, which is further reacted with **S1** from protonated **1a** with or without TFA to provide the C8-alkylated intermediate **S2**. The aminium radical intermediate **S2** via direct HAT event with the sulfate

**Table 1 | Optimization of reaction conditions[a]**

| Entry | Et-X | Additives | Yield[b] |
|---|---|---|---|
| 1[c] | **2a'** | none | 23 |
| 2 | **2a'** | none | 37 |
| 3 | **2a'** | Catechol | 90 |
| 4 | **2a** | Catechol | 87 |
| 5 | **2a** | Phenol | 32 |
| 6 | **2a** | Hydroquinone | trace |
| 7 | **2a** | Ethylene glycol | 27 |
| 8 | **2a** | none | 18 |
| 9 | **2a** | Catechol without $(NH_4)_2S_2O_8$ | 0 |
| 10 | **2a** | Catechol without TFA | 74 |
| 11 | **2a''** | Catechol with $MeB(OH)_2$ | 88 |
| 12 | **2a''** | Without Catechol | 10 |
| 13 | **2a''** | Catechol without $MeB(OH)_2$ | 0 |

[a]Reaction condition: Guanosine **1a** (0. 1 mmol), **2** (4.0 equiv.), photocatalyst (5 mol%), $(NH_4)_2S_2O_8$ (2.0 equiv.), additive (1.0 equiv.) and TFA (1.0 equiv.) in a solution of MeCN: $H_2O$ (1: 1, 0.1 M) under irradiation of 85 W white light at RT for 16 h.
[b]Yields were determined by LC-MS.
[c]2.0 equiv. ethyl potassium trifluoroborate was used instead of 4.0 equiv.

anion, delivered the nucleoside intermediate **S3**. Finally, the intermediate **S3** via subsequent deprotonation delivered the target alkylated nucleoside product **3a**.

**Reaction scope**

With the optimized reaction conditions in hand, we set out to study the scope of the reaction concerning alkyl boron radical precursors, including $RB(OH)_2$, $RBF_3K$, and RBPin (Fig. 2). Due to the inaccessibility of some $RB(OH)_2$ and $RBF_3K$, different radical sources were tested. Gratifyingly, various primary alkyl radical precursors with different lengths were found to be suitable substrates, providing the desired products (**3a**–**3g**) in yields of 50-83%. Notably, the yields of products (**3a** and **3c**) were much lower in the absence of catechol, which further indicates that catechol is necessary for this reaction.

Alkylboronic acids and derivatives possessing diverse functional groups such as alkenyl (**2h**), aromatic scaffolds (**2i** and **2j**), *tert* butoxy substituted (**2k**), ester (**2l**), alkynyl (**2m**), and azide (**2n**) could smoothly participate in this C8 alkylation, delivering the corresponding products (**3h**-**3n**). Notably, the products **3m** and **3n** with an alkyne or azide handle could be further functionalized via click reaction for biological application. As seen in **3o**-**3u**, the alkylation reactions of secondary alkylboronic acid derivatives typically proceeded well in good to excellent yield under standard conditions. Acyclic substituted alkylboronic acids afforded the corresponding alkylated products (**3o** and **3p**) in the yields of 80% and 72%, respectively, whereas **3p** was obtained as a mixture of diastereoisomers (d.r. = 1:1). The presence of cyclic structures, including cyclobutane (**3q**), cyclopentane (**3r**), cyclohexane (**3s**), oxane (**3t**), and piperidine (**3u**) rings, were found to

be amenable to deliver the desired products in 50–80% yields. However, no desired product **3v** was obtained in the case of the cyclopropyl system, probably due to the destabilization of the resulting cyclopropyl radical[38]. Additionally, we were pleased to find that the tertiary alkylboronic acids and derivatives gave desired products (**3w**-**3z**) in good to excellent yields (56–82%). It should be mentioned that this reaction cannot tolerate $MeB(OH)_2$, $ArB(OH)_2$, and other alkyl boron radical precursors (**3-1-3-4**, see Supplementary Table 13), which are also challenging substrates in this type of Minisci reactions[33].

Subsequently, a wide array of purine-based nucleosides was then explored (Fig. 3). Purine nucleosides with artificial ribose containing 2-fluoro, OMe, and O-methoxyethyl (MOE) substitution on the ribose moiety which were widely used in antisense oligonucleotides (ASOs) and small interfering RNAs (siRNAs) drug discovery[39–41] proceeded well to give the corresponding C8-alkylated products (**4b**-**4d**) with high yields. Satisfying results were also obtained from Ac protected substrate (**4e**, 72%); Due to its acid-sensitivity, 2′,3′-O-isopropylidene guanosine **3f** afforded the desired product in good yield (**4f**, 64%) in the absence of TFA. However, no desired product **4g** was obtained using deoxyguanosine (dG) as a substrate, resulting in the ribose-cleaved byproduct **4g'**. We reason that the glycosidic bond in dG is less stable than G under this condition, as radical hydrogen atom transfer (HAT) may take place at the 2′ position of ribose ring[42]. Decreasing the reactivity of ribose ring with the electron-withdrawing group could avoid the cleavage of the ribose. Fortunately, Ac group-protected deoxyguanosine (**1h**) was found to be a competent reaction partner for the current alkylation reaction, providing **4h** in high yield. The broad substrate scope and promising functional group compatibility of this

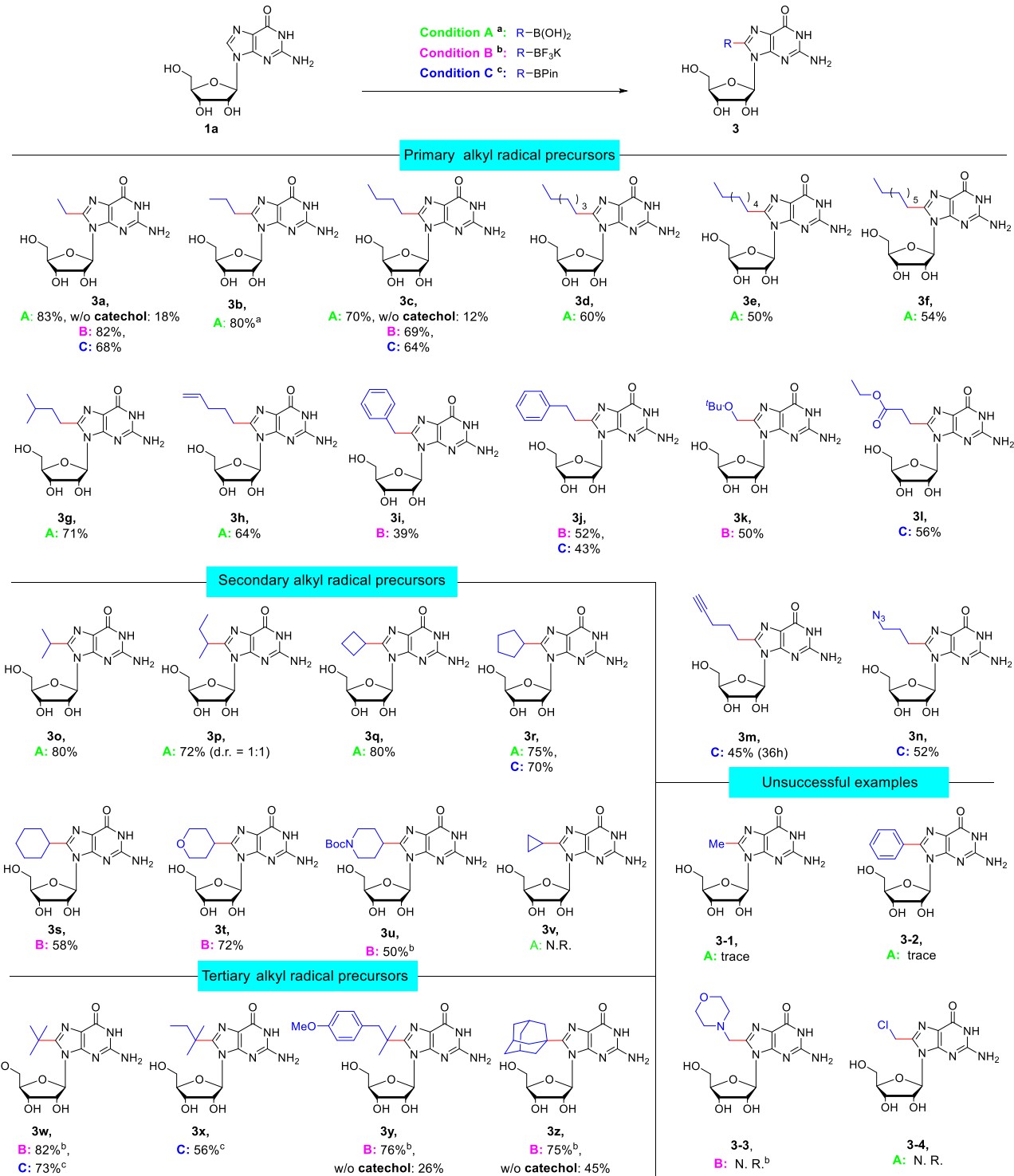

**Fig. 2 | Substrate scope of alkylboronic acids and derivatives.** [a]Condition A: Guanosine (1.0 equiv.), alkylboronic acid (4.0 equiv.), MesAcr (5 mol%), $(NH_4)_2S_2O_8$ (2.0 equiv.), TFA (1.0 equiv.) and catechol (1.0 equiv.) in MeCN: $H_2O$ (1: 1, 0.1 M) on 0.2 mmol scale; irradiated by 85 W white light at RT for 16 h; isolated yield. [b]Condition B: Alkyltrifluoroborate (2.0 equiv.) was used instead of alkylboronic acid. [c] Condition C: alkylpinacolyl boronate ester (2.5 equiv.) was used instead of alkylboronic acid, adding methylboronic acid (5.0 equiv.). Boc, t-Butyloxy carbony.

reaction encouraged us to evaluate their utility for late-stage modification of purine-based drug molecules, such as Penciclovir, Ganciclovir, Acyclovir (anti-HSV agents)[43] and Valganciclovir (anti-HIV agents) all progressed well to give the corresponding alkylation products with good to high yields (**4i-4l**, 52–90%). Meanwhile, alkylation of adenosine analogs was also investigated; however, the reactivity of simple adenosine is much lower than the G, providing mixed products with poor selectivity between the C2 and C8 positions. 2-fluoro- and 2-chloro substitutions blocking the C2 position were evaluated, and both gave products with good yields (**4n** and **4o**, 52% and 61%). However, no alkylation product **4p** was generated when employing C2-$NH_2$ substituted adenosine analog.

With this radical alkylation method in hand, we turned our attention to the C5 alkylation of uridine. C5 alkylated uridines are also

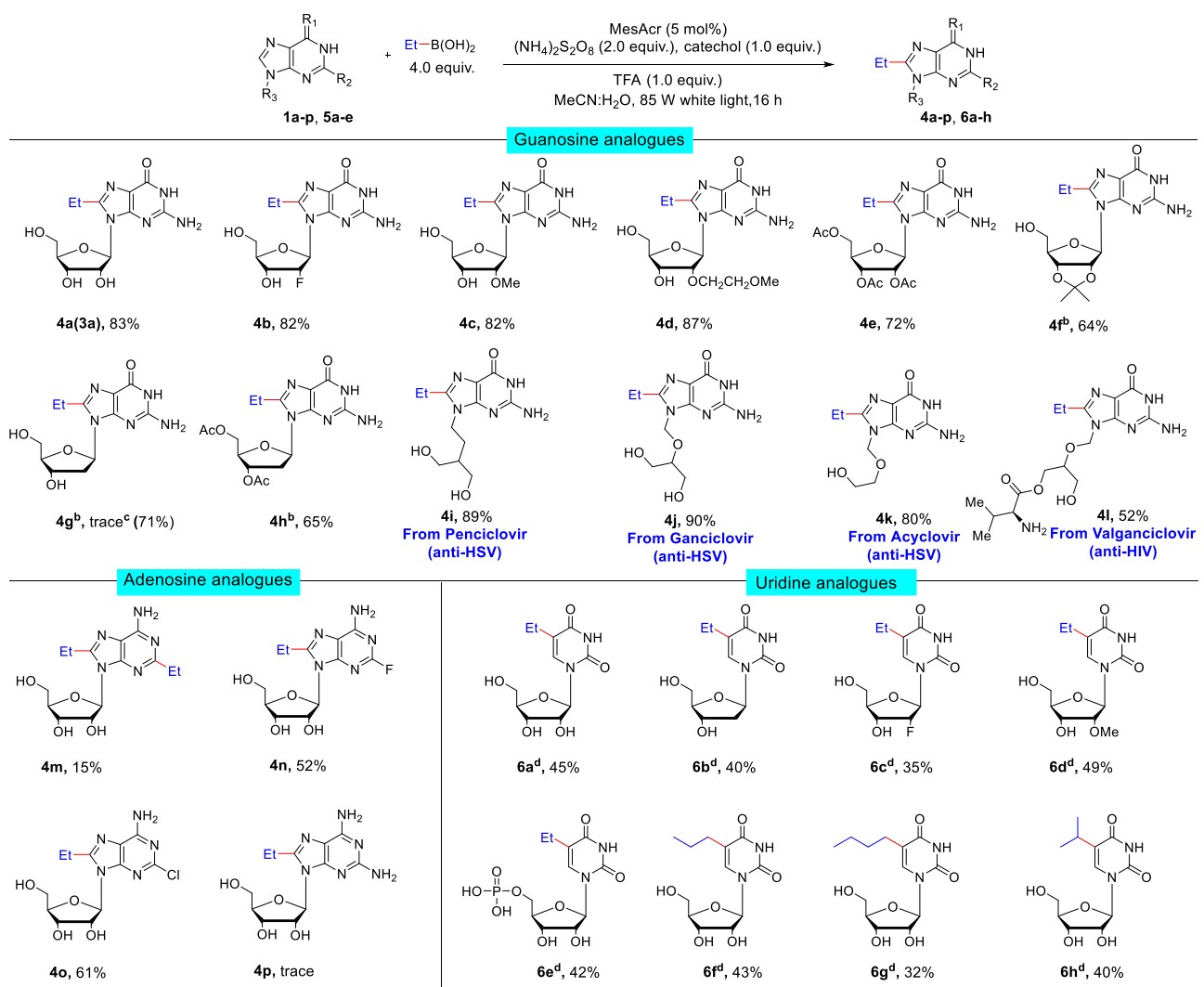

**Fig. 3 | Substrate scope of the nucleoside.** [a]General condition: Guanosine (1.0 equiv.), alkylboronic acid (4.0 equiv.), MesAcr (5 mol%), (NH₄)₂S₂O₈ (2.0 equiv.), TFA (1.0 equiv.) and catechol (1.0 equiv.) in MeCN: H₂O (1:1, 0.1 M) on 0.2 mmol scale; irradiated by 85 W white light at R.T. for 16 h; isolated yield. [b]No TFA was used. [c]Ribose-cleaved byproduct 4 g′ was obtained in 71% yield. [d]Uridine (1.0

equiv.), alkylboronic acid (4.0 equiv.), (NH₄)₂S₂O₈ (2.0 equiv.), MgCl₂ (2.0 equiv.) and catechol (1.0 equiv.) in DMSO: H₂O (1:1, 0.1 M) on 0.2 mmol scale; irradiated by 36 W blue LED at RT for 24 h. HSV herpes simplex virus, HIV human immunodeficiency virus.

important due to their application in drug development[44], such as C5-Et substituted drug molecular Edoxudine. Although various biological C5 alkylated uridines were prepared based on classic methods[45–48] (see Supplementary Fig. 8), direct C5 alkylation of uridine is still desirable due to the high-efficiency transformation. However, only a 3% yield of the desired product **6a** was obtained under the standard condition (see Supplementary Table 9). Then, dimethyl sulfoxide (DMSO)/H₂O solvent system improved the yield to 40%, and additive MgCl₂ instead of TFA further improved the yield to 45%. As shown in Fig. 3, a range of uridine nucleoside analogs was then investigated, affording the desired product (**6a-6e**) in moderate yields under the optimized condition. Alkyl chains carrying an ethyl, propyl, or isopropyl group can be successfully installed at the C5 site of the uridine moieties in moderate yields (**6f-6h**, 32-49%).

Having established the applicability of our methodology to a variety of purine nucleosides, we turned our attention to the modification of complex nucleotides (Fig. 4). Because of their lipophilicity, metabolic stability, binding selectivity, and bioabsorption characteristics, dinucleotide motifs are widely used in pharmaceuticals for antivirus and anti-tumor drug development[49–54]. Moreover, dinucleotide (UpG) was identified as an efficient "initiator dinucleotide" RNA

polymerases at the 5′ terminus of a transcript[54]. However, limited examples of late-stage functionalization of dinucleotides were reported due to the reactivity and selectivity issues. To our delight, the protocol has been effective in a range of dinucleotide scaffolds (Fig. 4a). Notably, alkylation of dinucleotides selectively occurred at the C8 site of the guanine subunit to give the desired alkylated products in good to excellent yields (**8a-8e**, 41–80%), without effecting other bases (uracil, thymine, cytosine, and adenine). For example, GpU and GpA dinucleotide with multiple reactive sites provided the single C8-Et desired products **8a** and **8d**. We reason that G is much more reactive than U and A in the nucleoside (**4a** *vs* **4m** in Fig. 3), as single U and A afforded trace or less product, so the selectivity could be achieved in these dinucleotide substrates. Meanwhile, dinucleotide holding two guanine subunits (GpG, **7e**), was selectively alkylated at both the C8 sites of guanine in 41% yield. Dinucleotides or oligonucleotides containing phosphorothioate (PS) linkage, with better chemical and biological stability[55] provided the corresponding products **8f** and **8f′** in good yields. Moreover, the protocol is also applicable to dGpT and dTpG, providing the desired product **8g** and **8h**, despite the fact that unprotected dG decomposed under this condition. We reason that the phosphate group has an electron-withdrawing effect similar to

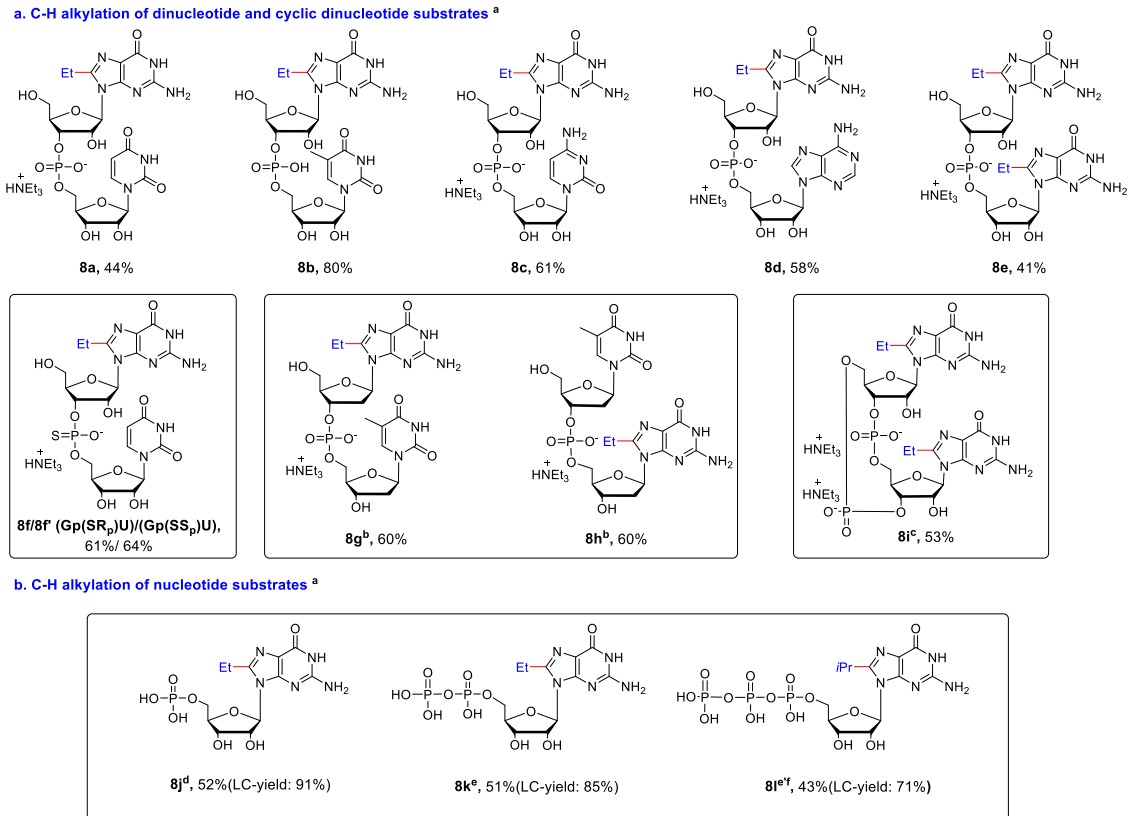

**a. C-H alkylation of dinucleotide and cyclic dinucleotide substrates** [a]

**8a**, 44%   **8b**, 80%   **8c**, 61%   **8d**, 58%   **8e**, 41%

**8f/8f' (Gp(SR_p)U)/(Gp(SS_p)U),** 61%/ 64%   **8g**[b], 60%   **8h**[b], 60%   **8i**[c], 53%

**b. C-H alkylation of nucleotide substrates** [a]

**8j**[d], 52%(LC-yield: 91%)   **8k**[e], 51%(LC-yield: 85%)   **8l**[e,f], 43%(LC-yield: 71%)

**Fig. 4 | Substrate scope of C-H alkylation of complex nucleotide substrates.**
**a** C-H alkylation of dinucleotide and cyclic dinucleotide substrates. [a]General condition: dinucleotides or nucleotides (1.0 equiv.), alkylboronic acid (4.0 equiv.), MesAcr (5 mol%), $(NH_4)_2S_2O_8$ (2.0 equiv.), TFA (1.0 equiv.) and catechol (1.0 equiv.) in MeCN: $H_2O$ (1: 1, 0.1 M) on 0.06 mmol scale; irradiated by 10 W blue LED at -10 °C for 24 h; isolated yield. [b]No TFA was used, and the reaction time was shortened to 5 hours. [c]Run on 0.04 mmol scale. **b** C-H alkylation of nucleotide substrates. [d]Irradiated by 36 W blue LED at r.t. for 16 h. [e]Irradiated by 85 W white light at r.t. for 16 h. 10 mM $NH_4HCO_3$ (aq.) was used instead of $H_2O$, and No TFA was used. [f]$i$-PrB(OH)$_2$ was used instead of EtB(OH)$_2$ as C8-Et-GTP cannot be separated from GTP.

Ac group (**4h** in Fig. 3), making the glycoside bond of deoxyguanosine more stable under this condition. Since modified cyclic dinucleotides (CDNs) are promising stimulators of interferon genes (STING) receptor agonizts in cancer immunotherapy[56], we also applied our method for late-stage modification of C-di-GMP, and di-alkylated C-di-GMP **8i** was successfully obtained in 53% yield. These promising results indicate that this method could be applied in the late-stage C8–H alkylation of G/dG in the RNA/DNA oligonucleotides.

Nucleotides, such as 5′-mono- and 5′-polyphosphorylated nucleosides play important functional and structural roles in biological systems[4,5]. Direct modification of nucleotides remains rare as nucleoside 5′-phosphate, including triphosphates, are stable only in slightly alkaline solution (pH ≈ 8), and very sensitive under acidic conditions. As seen in Fig. 4b, guanosine monophosphate (GMP) provided the desired product **8j** in good yield (LC-MS yield 91%, isolated yield 52%). However, guanosine diphosphate (GDP), and GTP afforded the corresponding C8-alkylated products (**8k** and **8l**) with low yield, along with phosphodiester bond hydrolyzed byproduct under the standard condition due to the acidic reaction condition. Since 5′-polyphosphorylated nucleosides is not stable in the acidic system, reaction condition without TFA may improve the yield. However, no better results were obtained in the absence of TFA, as this system after reaction is still very acidic (the pH value was 1.97, see Supplementary Table 12). Finally, this reaction proceeded well with moderate to good yields using the 10 mM $NH_4HCO_3$ as solvent (pH value was 2.72 after reaction, see Supplementary Table 12). In this condition, regrettably, we did not find a satisfactory method to prevent hydrolysis of **8 l**. It is worth noting that this method requires only one-step to obtain GTP

analogs with C8 substitution, which efficiently inhibits FtsZ polymerization, in comparison, the previous method required four steps[17].

## Applications of the methodology
To verify the practicability of this reaction manifold, the reaction of compounds **1a** and **2a** was scaled up to 6 mmol, and the desired product **3a** was obtained in 90% yield (1.6 g) after 24 h, opening the possibility of achieving a successful scale-up for further application (Fig. 5a). Moreover, **3a** was further alkylated at the site of C2-NH$_2$ by reductive amination, which is already applied for the modification of oligonucleotides[29]. It was able to obtain the target compound **10** in an excellent yield of 81%. Meanwhile, the reductive amination at the N2 site afforded alkylated **9**, followed by our C8-H radical reaction resulted in a high yield of the double alkylation product **10**. Modified nucleoside acids play an important role in the drug development and bio-imaging field. The compound **3l** with an alkyne handle could be bioconjugated with Zidovudine[57] which has an azide group via click reaction[58], resulting in the formal dinucleotide **11** in high yield (Fig. 5b). Additionally, compound **3m** also can be conjugated with coumarin **12** having azide group, affording the fluorescent compound **13** in 81% yield. In the same way, compound **3n** with an azide handle was successfully conjugated with biotin-alkyne **14** via click reaction.

## Alkylation of RNA/DNA oligonucleotides
After the successful application of this method to the C8-alkylation of single G and various dinucleotides, our attention then focused on the late-stage functionalization of RNA and DNA oligonucleotides (Tables 2 & 3). Although high selectivity between the G base and other

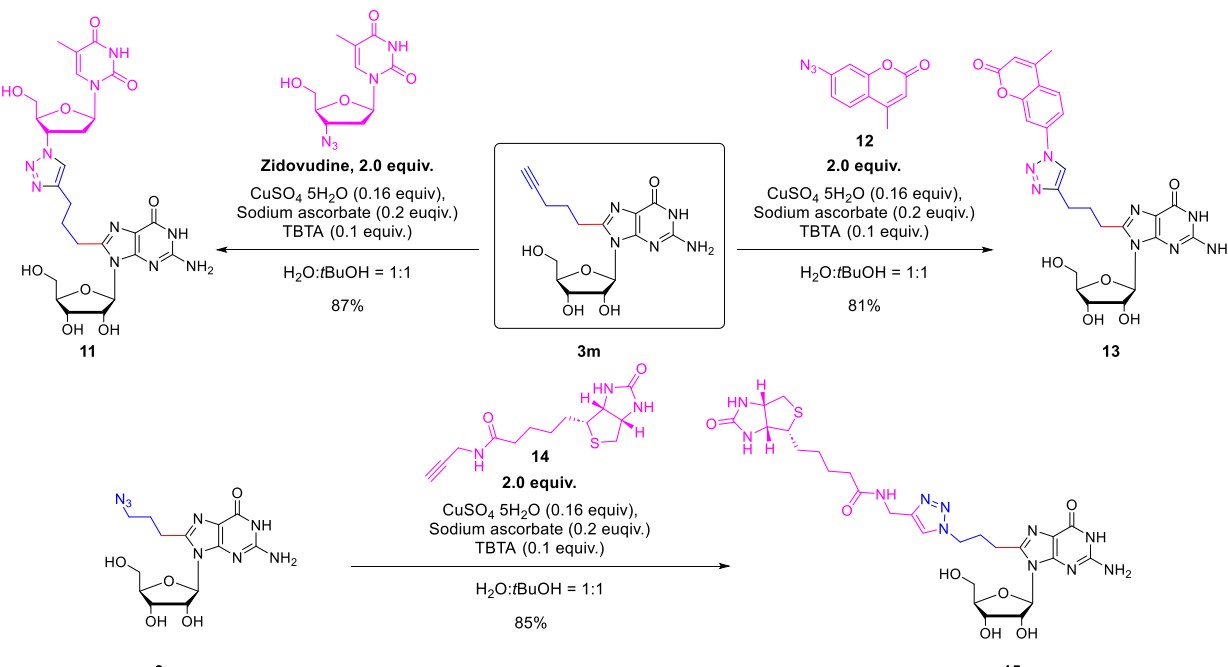

**Fig. 5 | Applications of the methodology. a** Sequential functionalization of G with radical alkylation and reductive amination reaction. **b** Bioconjugation with 3 m and 3n via click reaction. *i*Pr isopropyl, TBTA Tris((1-benzyl-4-triazolyl)methyl)amine.

bases was demonstrated in the dinucleotides system (Fig. 4a), the reactivity problem may exist as the substrate concentration of oligonucleotides reaction was much dilute than the normal reaction condition (mM vs M)[59]. Indeed, no desired product was obtained using the guanosine as a substrate under the condition of normal concentration of the oligonucleotides (100 nmol, 0.2 mM) (see Supplementary Table 17, **entry 1**). By simultaneously increasing the equivalents of ethylboronic acid, catechol, $(NH_4)_2S_2O_8$, TFA, and MesAcr (see Supplementary Table 17, **entries 2-4**), a much better result was obtained with a yield of 89% for the target product under the condition with 400 times of above reagents compared with the original condition). Due to the stability of oligonucleotides under relatively high concentration TFA and solubility in the water system, the equivalents of the aforementioned compounds were reduced while still keeping the reactivity (see Supplementary Table 17, **entries 5-21** for full optimization). Finally, quantitative yield (99%) was achieved with the full conversion of starting material under the condition of alkylboronic acid (400.0 equiv.), MesAcr (50 mol%), $(NH_4)_2S_2O_8$ (200.0 equiv.), TFA (50.0 equiv.), and catechol (100.0 equiv.) in MeCN/$H_2O$ (0.25 mL/0.25 mL) (see Supplementary Table 17, **entry 10**, Table 2, **entry 1**). Notably, catechol was demonstrated to be essential for this reaction (see

Supplementary Table 17, **entry 18**). Furthermore, several dinucleotides containing with G also proceeded well under this dilute reaction (Table 2, **entries 2-4**), affording the mono C8-alkylation of G products despite using the excess of ethylboronic acid (400 equiv.). With this optimized methodology in hand, we set up to test the application of RNA oligonucleotides. Firstly, **ONs 1-3** (6 nt) with a guanine base in different sites (internal, or 5′, 3′ terminal position) and without adenine base in the sequence delivered the desired product in high yield (Table 2, **entries 5-7**), and the product and reactive site were confirmed by high-resolution mass spectrometry (HRMS), matrix-assisted laser desorption/ionization Fourier transform mass spectrometry (MALDI-FTMS) (see Supplementary Figs. 87–91) and nuclease digestion method (see Supplementary Figs. 92 and 93). Then, **ON 4** containing both guanine and adenine also proceeded with high selectivity for C8 guanine in 78% yield, and no poly-substituted products were observed (Table 2, **entry 8**). **ON 5** switching the position of A and G also afforded the desired product in high yield (Table 2, **entry 9**). **ONs 6-8** with 7-9 nt resulted in good yields (Table 2, **entries 10-12**). Finally, different alkyl groups, such as *n*-Pr, azide chain, cyclobutyl, and cyclopentyl, were successfully introduced to the C8 G in the **ON 5** (Table 2, **entries 13-16**). Since there are rare examples of C8-alkylated

**Table 2 | Yields & MALDI-FTMS and ESI-QToF analysis of C8–H alkylation of oligonucleotides [a]**

| Entry | Oligonucleotides & sequence | Yield [%][b] | Free radical precursors | Calcd. exact mass | Exptl. m/z (MALDI-FTMS) |
|---|---|---|---|---|---|
| 1 | **Guanosine** G | 99[c] | Et-B(OH)$_2$ | 311.1230 | [M-H]$^-$ = 312.1309[d] |
| 2 | **GpA** 5'-GA-3' | 71[c] | Et-B(OH)$_2$ | 640.1755 | [M+Na]$^-$ = 663.1656[d] |
| 3 | **GpG** 5'-GG-3' | 59[c] | Et-B(OH)$_2$ | 684.2017 | [M-H]$^-$ = 683.1948[d] |
| 4 | **GpC** 5'-GC-3' | 66[c] | Et-B(OH)$_2$ | 616.1643 | [M-H]$^-$ = 615.1554[d] |
| 5 | **ON 1** 5'-GUUUCC-3' | 78 | Et-B(OH)$_2$ | 1839.2814 | [M + H]$^+$ = 1840.31214 |
| 6 | **ON 2** 5'-UUUGCC-3' | 71 | Et-B(OH)$_2$ | 1839.2814 | [M + H]$^+$ = 1840.28383 |
| 7 | **ON 3** 5'-UUUCCG-3' | 80 | Et-B(OH)$_2$ | 1839.2814 | [M + H]$^+$ = 1840.29030 |
| 8 | **ON 4** 5'-GUUACC-3' | 78 | Et-B(OH)$_2$ | 1862.3087 | [M + H]$^+$ = 1863.31278 |
| 9 | **ON 5** 5'-AUUGCC-3' | 69 | Et-B(OH)$_2$ | 1862.3087 | [M + H]$^+$ = 1863.31867 |
| 10 | **ON 6** 5'-GCUAUCU-3' | 63 | Et-B(OH)$_2$ | 2168.3340 | [M + H]$^+$ = 2169.36146 |
| 11 | **ON 7** 5'-GUUACCUU-3' | 73 | Et-B(OH)$_2$ | 2474.3593 | [M + H]$^+$ = 2475.35981 |
| 12 | **ON 8** 5'-GUUACCUCU-3' | 74 | Et-B(OH)$_2$ | 2779.4006 | [M + H]$^+$ = 2780.42431 |
| 13 [e] | **ON 5** 5'-AUUGCC-3' | 59 | $n$Pr-B(OH)$_2$ | 1876.3243 | [M + H]$^+$ = 1877.31866 |
| 14 [f] | **ON 5** 5'-AUUGCC-3' | 31 | N$_3$(CH$_2$)$_3$-Bpin | 1917.3257 | [M + H]$^+$ = 1918.30266 |
| 15 [g] | **ON 5** 5'-AUUGCC-3' | 57 | Cyclobutyl-B(OH)$_2$ | 1888.3243 | [M + H]$^+$ = 1889.30149 |
| 16 [h] | **ON 5** 5'-AUUGCC-3' | 63[i] | Cyclopentyl-B(OH)$_2$ | 1902.3400 | [M + H]$^+$ = 1903.33673 |

[a]General condition: Oligonucleotide (100 nmol), ethylboronic acid (400.0 equiv.), MesAcr (50 mol%), (NH$_4$)$_2$S$_2$O$_8$ (200.0 equiv.), TFA (50.0 equiv.), and catechol (100.0 equiv.) in MeCN (0.25 mL), H$_2$O (0.25 mL), irradiated by 85 W white light at r.t. for 16 h.
[b]Yields were determined by LC-MS and used the analytical method B.
[c]Yields were determined by LC-MS and used the analytical method A.
[d]The mass data was collected by using HRMS.
[e]$N$-butylboronic acid (400.0 equiv.) was used, and reaction time was extended to 24 hours.
[f]alkylpinacolyl boronate esters (400.0 equiv.) was used, and methylboronic acid (400.0 equiv.) was additionally added.
[g]Cyclobutylboronic acid (200.0 equiv.) was used, equivalent of catechol has been reduced to 200, and irradiated by 10 W white LED at 10 °C.
[h]Cyclopentylboronic acid (400.0 equiv.) was used, and irradiated by 10 W white LED at 10 °C.
[i]Yields were determined by LC-MS and used the analytical method C.

oligonucleotides that have been reported, these various C8-alkylated oligonucleotides may have applications in the area of nucleic acid drugs, such as ASO, siRNA, and aptamer.

Meanwhile, dinucleotide dGpT and dTpG could also proceed well under the modified condition [short reaction time (5 h), lower temperature (0 °C), and without TFA] (Table 3, **entries 1, 2**). For this protocol, the aforementioned condition is needed to avoid the glycoside bond breaking in the reaction, since ethylboronic acid, catechol, (NH$_4$)$_2$S$_2$O$_8$, TFA, and MesAcr are far excess. Then, DNA oligonucleotides with different sequences and lengths were also tested. **ONs 9** and **10** with G in the 5' and 3' sites of the sequence, the generation of by-products was observed, and the yield of the target product was relatively less (Table 3, **entries 3, 4**). While **ON 12** with internal dG in the sequences provided the desired product in 80% yield with staring material recovered (Table 3, **entry 6**). Gratefully, **ONs 11, 13-17** with different sequences and lengths afforded the desired product in good yields (Table 3, **entries 5, 7-11**).

We have carried out the experiment with DNA oligonucleotides **ON 18** (5'-dTGTCGC-3') containing two guanines as substrate (Table 4, **entry 1**). It was found that monosubstituted product **16a** and disubstituted product **17a** can be obtained with yields of 33% and 67%, respectively. Next, we used RNA oligonucleotides containing two guanines and one adenine (5'-CGAUGU-3') as substrates (Table 4, **entry 2**). The results showed two monosubstituted products **16b** were generated, with a total yield of 33%, and a disubstituted product **17b** was generated, with a yield of 56%.

Previous late-stage cross-coupling reactions on oligonucleotides strongly depend on the sequence and secondary structure, showing the differences between internal and terminal position and single- and double-stranded DNA[60–62]. From our preliminary results (see Supplementary Table 22 and Supplementary Fig. 82 for more detail), both chains were found to have ethylated products generated. The G on these two strands is in no special position, both in internal and terminal position, which might mean that this method can be directly used for

## Table 3 | Substrate scope of C–H ethylation of the ssDNA oligonucleotides[a]

| Entry | Oligonucleotides & sequence | Yield [%][c] | Calcd. exact mass | Exptl. m/z (MALDI-FTMS) |
|---|---|---|---|---|
| 1[b] | dGpT<br>5'-dGT-3' | 70[d] | 599.1741 | [M + H]+ = 600.1807[e] |
| 2[b] | dTpG<br>5'-dTG-3' | 65[d] | 599.1741 | [M + 2Na]+ = 644.1451[e] |
| 3 | ON 9<br>5'-dGTTCC-3' | 43 | 1481.3129 | [M + H]+ = 1482.32446 |
| 4 | ON 10<br>5'-dTTCCG-3' | 33 | 1481.3129 | [M + H]+ = 1482.32241 |
| 5 | ON 11<br>5'-dCGTT-3' | 63 | 1192.2665 | [M + H]+ = 1193.27804 |
| 6 | ON 12<br>5'-dTTGCC-3' | 80 | 1481.3129 | [M + H]+ = 1482.32368 |
| 7 | ON 13<br>5'-dCATGT-3' | 64 | 1505.3241 | [M + H]+ = 1506.34015 |
| 8 | ON 14<br>5'-dCCCGTTT-3' | 62 | 2074.4053 | [M + H]+ = 2075.40114 |
| 9 | ON 15<br>5'-dCACGTTT-3' | 69 | 2098.4165 | [M + H]+ = 2099.43403 |
| 10 | ON 16<br>5'-dCCTTGTTCC-3' | 51 | 2667.4977 | [M + H]+ = 2668.47158 |
| 11 | ON 17<br>5'-dCACTTGTTC-3' | 59 | 2691.5089 | [M + H]+ = 2692.49906 |

[a]General condition: oligonucleotide (100 nmol), ethylboronic acid (400.0 equiv.), MesAcr (50 mol%), $(NH_4)_2S_2O_8$ (200.0 equiv.), TFA (50.0 equiv.), and catechol (100.0 equiv.) in MeCN (0.25 mL), $H_2O$ (0.25 mL), irradiated by 85 W white light at r.t. for 16 h.
[b]The reaction was irradiated by 10 W LED and carried out at 0 °C for 5 h. [c]Yields were determined by LC-MS and used the analytical method B. [d]Yields were determined by LC-MS and used the analytical method A. [e]The mass data was collected by using HRMS.

modifying dsDNA without being affected by its pairing. However, the cosolvent and lack of salts may not support double-stranded hybridization, and it is possible that the strands are reacting in single-stranded form.

Since click chemistry has been widely used in nucleic acids for labeling, ligation, cyclization and other applications. We also conducted the application of modified oligonucleotide via click chemistry (Fig. 6). Firstly, an oligonucleotide (18) with azide group was obtained from ON 5 in 31% yield, after separation with pre-HPLC, followed by click reactions to conjugate a biotin-alkyne 14 to afford the oligonucleotide 19 in 41% yield.

## Discussion

Modified nucleosides, nucleotides, and oligonucleotides are important in the fields of medicinal chemistry, chemical biology, biotechnology, and nanotechnology. The introduction of functional groups into the nucleobases of these molecules mostly relies on the laborious de novo chemical synthesis, and late-stage functionalization of nucleosi(ti)des and oligonucleotides have been developed in recent years. There are successful examples of C8-arylation, alkenylation, and alkynlation of purines of oligonucleotides using a cross-coupling approach. In contrast, there are rare methods for selective alkylation at the C8 site of G in RNA/DNA oligonucleotides, which prevents further application of these alkylated nucleosi(ti)des and oligonucleotides. Here, we have achieved C8-alkylation of nucleosides, nucleotides, and oligonucleotides via mild Minisci radical reaction. We believe that the introduction of various alkyl groups at the C8 position of guanine is a

## Table 4 | Substrate scope of C–H ethylation of the oligonucleotides containing two guanines[a]

| Entry | Oligonucleotides & sequence | Yield of 16/17[%][b] | Calcd. exact mass | Exptl. m/z (MALDI-FTMS) |
|---|---|---|---|---|
| 1 | ON 18<br>5'-dTGTCGC-3' | 33[c]/67[d] | 1810.3654[c]/1838.3967[d] | [M + H]+ = 1811.37688[c]/1839.41817[d] |
| 2 | ON 19<br>5'-CGAUGU-3' | 33[c]/56[d] | 1902.3148[c]/1930.3461[d] | [M + H]+ = 1903.31515[c]/1931.34806[d] |

[a]Condition: oligonucleotide (100 nmol), ethylboronic acid (400.0 equiv.), MesAcr (50 mol%), $(NH_4)_2S_2O_8$ (200.0 equiv.), TFA (50.0 equiv.), and catechol (100.0 equiv.) in MeCN (0.25 mL), $H_2O$ (0.25 mL), irradiated by 85 W white light at r.t. for 16 h. [b]Yields were determined by LC-MS and used the analytical method B. [c]Data of monosubstituted product(s). [d]Data of disubstituted product.

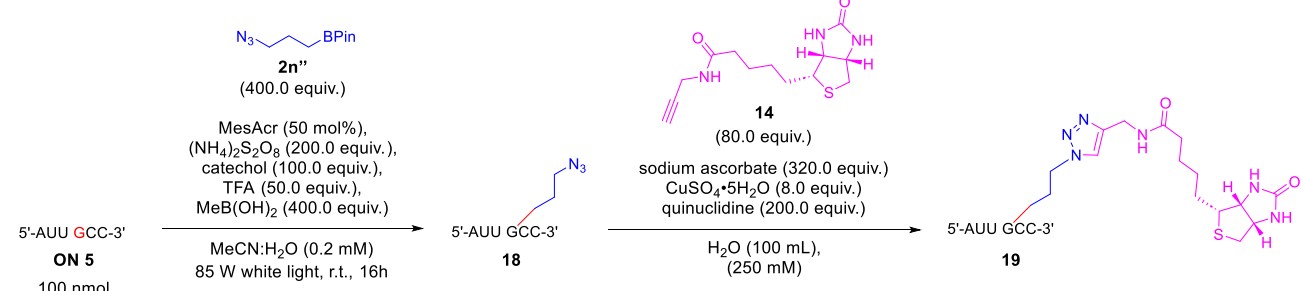

**Fig. 6 | Click functionalization of C8-H alkylation for oligonucleotide.** The alkylation of alkyne handle in oligonucleotide functionalization and subsequent elaboration by Huisgen cycloaddition.

good supplement to the previous aryl/alkenyl/alkynyl substitution, these alkylated nucleosi(ti)des and oligonucleotides will have potential application in the field of nucleic acid.

With our methods, a wide array of readily accessible primary, secondary, and tertiary alkylboronic acids and derivatives were successfully introduced to the C-8 unprotected guanine and analogs in just one-step (Figs. 1 and 2). In addition, C5 alkylation of uridine was also achieved in moderate yield via modified methods (Fig. 2). The limitation of our method is that methyl and phenyl groups cannot be introduced using current conditions. We believe that this practical alkylation method for the modification of nucleosides will be useful in medical chemistry, especially for medicinal chemists to quickly set up nucleoside compound libraries for drug discovery.

Our method also demonstrates its application for the late-stage modification of nucleotides, such as dinucleotides and CDNs. Since de novo synthesis of CDNs requires multi-steps, setting up CDNs libraries for drug screening is time and resource-consuming. Our method can directly modify the CDN (like C-di-GMP in Fig. 4), and has the potential to quickly provide a CDNs library for screening. Moreover, our methods can easily introduce alkyl group to the C-8 position of nucleoside triphosphates, which cannot be easily achieved using previous methods (like GTP in Fig. 4). C-8 modification will result in altering purine geometry to the nonstandard *syn* conformation, which strongly hinders its ability to undergo base pairing, only 8-substituted purine derivatives bearing small groups, like amino, bromo, or methyl are generally good substrates for DNA polymerases. In contrast, 8-Ph-dATP is too bulky for the polymerase to accept as a substrate[25]. Due to the hard accessibility of NTPs bearing other alkyl groups, fewer studies were reported on these functionalized oligonucleotides. After preparation of alkyl functionalized NTPs using our one-step method, we will study their compatibility with DNA/RNA polymerase and further application.

Since C and T are not reactive under this radical condition, meanwhile, U and A are much lower reactive compared with G, so selective late-stage modification of G in the RNA oligonucleotides was successfully achieved. Although simple dG is not compatible with the current condition due to the ribose-cleavage, 3' and 5' phosphate groups lower the electron density of the ribose ring to prevent the side reaction of ribose-cleavage, so direct alkylation of DNA oligonucleotides could also be accomplished. In the future, one major application of this method for the modification of DNA/RNA oligonucleotides is in the area of nucleic acid drugs, such as ASO, siRNA and aptamer. Due to the selectivity and compatibility with PS in the oligonucleotides, various alkyl groups could be introduced to these oligonucleotides containing less G (one or two), and further studies of the effects of these alkylated modifications on the stability, activity, and toxicity in nucleic acid drugs will be carried out. Although the selective function of G between different bases can be achieved in the oligonucleotides, the selective function of specific G in the sequence with many G is still challenging under the current method. If specific G in the oligonucleotides bearing many G (more than 3) need to be alkylated, solid-phase synthesis would be a more suitable approach using alkylated monomers. Another application of this method would be the bioconjugation of the oligonucleotides for labeling and cross-linking with other biomacromolecules[63].

In conclusion, we have successfully developed a visible light-mediated catechol-assisted direct C8 alkylation of the guanines in guanosine (together with its congeners), guanosine-containing dinucleotides, and oligonucleotides. This robust methodology can also be applied to the late-stage functionalization of drugs and some fragile biomolecules, such as GMP, GDP, and GTP. The realization of site-selective post-functionalization of 20 oligonucleotides indicates its potential application values in the areas of medicinal science, chemical biology, biotechnology, and nanotechnology other than organic chemistry.

## Methods

### General procedure for the C8−H alkylation of nucleoside

To a 10 mL Schlenk tube containing a Teflon stir bar was charged with nucleoside substrates (0.2 mmol, 1.0 equiv.), alkylboronic acids or alkyltriuoroborate (0.8 mmol, 4.0 equiv.), MesAcr (0.01 mmol, 0.05 equiv.), $(NH_4)_2S_2O_8$ (0.4 mmol, 2.0 equiv.), catechol (0.2 mmol, 1.0 equiv.), 1.0 mL $CH_3CN$, 1.0 mL $H_2O$, and trifluoroacetic acid (0.1 mmol, 1.0 equiv.) sequentially. The Schlenk tube was sealed with a rubber plug and taped, utilizing the freeze-pump-thaw (FPT) method for deaeration. The reaction system was exposed to an 85 W white light, monitoring the progress of the reaction by thin layer chromatography (TLC) (DCM/MeOH = 5/1) or liquid chromatography-mass spectrometry (LC-MS). The mixture was concentrated under reduced pressure, and the crude product was purified by silica gel flash column chromatography (C18 Spherical silica) using MeOH/$H_2O$ as an eluent to give the pure product.

### General procedure for the C8−H alkylation of complex nucleotide substrates and RNA/DNA Oligonucleotides

First, preparing aqueous solutions of the complex nucleotide substrates (0.01 mM), alkylboronic acids (4.0 mM), $(NH_4)_2S_2O_8$ (2.0 mM), and trifluoroacetic acid (0.5 mM), as well as the acetonitrile solutions of MesAcr (0.005 mM) and catechol (1.0 mM). Then, utilizing a suitable pipette gun, sequentially, to take 210 μL $H_2O$, 190 μL MeCN (ensuring the total amount of $H_2O$ and acetonitrile is 250 mL respectively), 10 μL of complex nucleotide substrates solution (0.10 μmol, 1 equiv.), 10 μL of alkylboronic acids solution (40 μmol, 400 equiv.), 10 μL of $(NH_4)_2S_2O_8$ solution (20 μmol, 200 equiv.), 20 μL of MesAcr solution (0.050 μmol, 0.5 equiv.), 40 μL of catechol solution (10 μmol, 100 equiv.), and 10 μL of trifluoroacetic acid solution (5.0 μmol, 50 equiv.) into a 5 mL Schlenk tube

containing a Teflon stir bar, sealed with a rubber plug, and taped, utilizing the FPT method for deaeration, liquid nitrogen for freeze, and argon as an inert gas (see Supplementary Information for specific analysis procedure). Pay attention to safety when using this system, and keep the Schlenk Line in the state of blowing argon when thawing, which is to maintain atmospheric pressure and an inert gas atmosphere. The time from mixing various solutions and solvents to freezing and vacuuming should not exceed ~10 minutes. Then, the reaction system was exposed to an 85 W white light. The yield and recovery of the reaction are detected by LC-MS (see Supplementary Information for specific analysis method). The operation steps for RNA/DNA Oligonucleotide are basically the same as those above, while diethyl pyrocarbonate-treated water (DEPC-treated Water) (RNA enzyme and DNA enzyme free) was used instead of $H_2O$, as well as dissolving the sub-packed RNA/DNA Oligonucleotide (100 nmol) with 90 μL (30 μL × 3 times) DEPC water.

### Reporting summary

Further information on research design is available in the Nature Portfolio Reporting Summary linked to this article.

## Data availability

Details about materials and methods, experimental procedures, mechanistic studies, and NMR spectra are available in the Supplementary Information.

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

## Acknowledgements

This work is supported by the Natural Science Foundation of Qinghai Province (2023-ZJ-917M, G.C.), the National Natural Science Foundation of China (NSFC) (Grant no. 22001165 and 22271186, G.C.) and Shanghai Jiao Tong University (SJTU). We thank professor Juntao Ye (SJTU) for the helpful discussion. We also thank Dr. Raghunath Bag (SJTU) and Dr. Karuppu Selvaraj (SJTU) for their editorial advice on manuscript preparation.

## Author contributions

Y.G. and G.C. conceived and directed the project. R.X., W.L., Y.Z., G.X., Q.Z., Y.H. and Y.L. performed the experiments and analyzed the data. G.C. wrote the manuscript with input from all authors.

## Competing interests

A patent application by G.C., Y.G., W.L., R.X., G.X. and Y.L. detailing part of this research was filed through the Patent Office of the People's Republic of China (June 2022). The other authors declare no competing interests.
