## [Peer Review File · Nature Communications]

REVIEWER COMMENTS

Reviewer #1 (Remarks to the Author):

This is an interesting and potentially useful work on C-H alkylation of nucleosides, nucleotides and oligonucleotides using photochemical Minisci reaction. It seems that for alkylation of guanine, the reaction can be done with number of alkylboronates and is quite general. It works also (less efficiently) with adenosine and uridine. It even works on nucleotides, dinucleotides and short oligonucleotides with preference for guanosine. The work is potentially acceptable for Nat. Commun. after revision - see below:

1. What is for me a bit difficult to believe is the reaction of GTP - in our hands the NTPs are not stable for 16 h in aqueous buffer - did the authors observe any hydrolysis of the triphosphate ?

2. For reactions of ONs - the authors have only MS analysis and they just assume the substitution proceeded on G (rather than on other nucleobases) - they should prove it at least in 1-2 cases

3. The authors should better discuss the outcome of the reactions of ONs - related cross-coupling reactions on ONs strongly depend on the sequence and secondary structure - see for example:

<https://doi.org/10.1002/cbic.202100608>

<https://doi.org/10.1021/acs.bioconjchem.0c00466>

<https://doi.org/10.1093/nar/gky185>

the authors should cite and discuss these papers - they show differences between internal and terminal position and between single- and double stranded DNA (basically cross-couplings do not proceed on dsDNA)

I highly recommend that the authors also try the C-H activation on dsDNA with G in internal and terminal position to verify if the transformation is suitable for modification of DNA

- cross-coupling reactions on cyclic-dinucleotides also should be cited:

<https://doi.org/10.1021/acs.jmedchem.2c01305>

Reviewer #2 (Remarks to the Author):

In this article, the authors report a new methodology for the facile, late-stage functionalization of the C8 position of guanine nucleosides, nucleotides, and oligonucleotides. To do so, the authors first optimized the conditions of a photo-mediated Minisci reaction and demonstrated that the addition of catechol promoted the formation of alkyl radicals and led to marked increases in yields. After this optimization step, the authors then explored the scope of the reaction by changing the nature of the alkyl radical precursors as well as the nucleoside. Finally, the authors demonstrate that this guanine functionalization reaction is also compatible with more complex substrates such as dinucleotides and to a certain extent nucleoside triphosphates and short, single-stranded oligonucleotides. All experiments were conducted in a competent manner, the characterization of all products is thorough, the supporting information is of very good quality, and the manuscript is clear. Overall, C8 functionalization of guanine nucleosides, nucleotides, and oligonucleotides is of interest in numerous fields and this method will be of broad appeal. Hence, publication in Nature Communications is recommended pending some revisions:

- While pyrimidine oligonucleotides are rather stable in TFA, this is much less the case for purine containing sequences. Usually, DNA oligonucleotides containing dA residues are depurinated and eventually hydrolyzed in high proportions upon incubation with TFA even for shorter periods of time (see e.g. 10.1080/15257770701508216; 10.1021/acs.analchem.9b03625). It is thus difficult to rationalize how ODNs 12, 14, and 16 can survive 17h incubation in the presence of 50 equivalents of TFA and yield the expected product in over 55% yields. A thorough investigation on the stability and integrity of dA containing DNA oligonucleotides needs to be provided to support these findings and conclusions.
- Page 7: PS bonds are known to be a redox site that can compete with G oxidation and be oxidized by a variety of radicals (see e.g. NAR 2019, 47, 11514). The authors need to provide an explanation for the absence of reactivity of the PS bonds under these reaction conditions.
- Page 7: usually, nucleoside triphosphates are highly unstable at pH below 3, especially when incubated for longer periods of time. The authors thus need to provide an HPLC or LCMS time course on this reaction to univocally demonstrate that the triphosphate moiety is not affected by these conditions.
- It is often desirable to functionalize the C-8 position of purines with functional groups such as amines, thiols, imidazoles, or carboxylic acids (see e.g. NAR 2009, 37, 1638; JACS 2004, 126, 12720; JACS 2021, 143, 18960; PNAS 2010, 107, 20720). The authors should evaluate the possibility of introducing such functional groups at position C-8 of guanine.
- Scheme S13-2 should probably be moved to the main manuscript since this is a rather important piece of information.
- Structure of compound 4g' should be provided in the ESI.
- The authors need to specify in each case how much (mg and mmoles) starting material were engaged in each reactions and not only provide final yields.
- A thorough discussion section needs to be included in this manuscript.

Reviewer #3 (Remarks to the Author):

The authors describe photo-mediated Minisci chemistry for modifying C-8 of guanosine and C-5 of uridine. For nucleosides and analogues the reaction proceeds reasonably well, especially for C-8 of guanine base in nucleosides. For dinucleotides and oligonucleotides the chemistry is not as efficient. Given the limited utility of C8 alkyl substitution in the latter (see below), I am not convinced of the high utility of this method beyond nucleoside modification.

1. In the introduction the authors have described prior chemistry for modification of C-H positions on nucleosides, including C-8 of guanine and C-5 of uracil. Among the most successful methods is the cited Suzuki chemistry, but as the authors rightly note, the Suzuki chemistry requires the use of a halogenated starting material. It is relatively simple (or inexpensive to purchase) such bromonucleosides, but it does add a step that the authors' method does not require. I believe this is the strongest point in favor of the new work, although there are some important limitations (see below).
2. In their introductory examples of late-stage functionalization of purines in DNA/RNA, the authors have missed two significant newer chemistries that deserve to be mentioned and cited. The first is the use of kethoxal and derivatives to react with guanine (PMID: 32015521) and a second is the use of SNAr chemistry to modify iodopurine (PMID: 35119264).
3. Figure 3 shows that reactions of uracil in nucleosides proceed with poor to moderate yields. For adenine nucleosides, the C2 position is also modified by the chemistry unless blocked by other substitution. Thus it is guanine C-8 modification that has the chief useful application here in my view.
4. Figure 2 shows the scope of alkyl groups that can be added at C-8 of guanosine. The scope is fairly broad for primary and secondary alkyl groups and provides moderate to good yields (39 to 82%). Looking at the substrate scope, the reaction proceeds relatively well for a variety of derivatives of guanosine and deoxyguanosine. It is carried out without need for protecting groups. This seems a useful addition to the field of nucleoside chemistry.
5. For dinucleotides and nucleoside phosphate derivatives (Fig. 4), the yields are somewhat lower, with several examples in the 40-60% range. Finally, Table 2 (entries 5-12,15-22) lists yields for RNA/DNA oligonucleotide modification. For RNA the yields are moderate to good (63-80%), while for DNA yields are somewhat lower (33-72%).
6. I am not convinced that this reaction method has broad utility for chemists who modify DNA/RNA. The authors note a number of possible applications of C-8 modification in the introduction, but these may not apply to their chemistry. For example, fluorescent nucleosides with modifications at C-8 require aryl

or alkenyl/alkynyl substitution to extend conjugation, and this is not possible with the new chemistry, which seems limited to alkyl substitution. This is a limitation of the new chemistry, since the earlier Suzuki chemistry allows for aryl and alkenyl substitution. One additional limitation of the new chemistry for RNA/DNA is that it appears to require strict inert atmospheres with freeze/thaw degassing and Schlenk apparatus. This also lowers the chances of broader adoption.

7. Second, C-8 modification of guanine is usually not very compatible with DNA/RNA biochemical function. Modifying the C-8 position results in altering purine geometry to the nonstandard syn conformation, which strongly hinders its ability to undergo base pairing. This also prevents it from being a good substrate for polymerase enzymes.

8. Finally, the oligonucleotides tested in Table 2 only contain one guanine base, thus guaranteeing reaction at one preferred site. However, this does not seem practically useful, as nearly all applications of DNA/RNA will require more than one guanine (usually many of them). Can the authors suggest a useful application where the chemistry could be helpful when there are several or many guanines in DNA or RNA? If they can make a strong argument for this, or demonstrate utility of this, it could add significantly to the importance and broad interest of this work.

9. In addition, post-synthesis modification of an oligo by the authors' approach will require HPLC purification to separate the product from the significant amount of unreacted starting material or other byproducts. Thus it seems more likely that their chemistry would have more value by simply modifying the nucleoside monomer and incorporating it into synthetic oligonucleotides by solid phase synthesis. By that method, nearly 100% modification would be achieved.

10. Finally, the authors only documented the addition of an ethyl group at C8 of G in RNA/DNA. If the utility in RNA/DNA is to be supported, then the authors should demonstrate reaction with a broader range of modifications, including ones that biochemists or, for example, siRNA medicinal chemists, might find useful.

11. In summary, I believe that this chemistry is a useful addition for making C-8 alkyl-modified guanine nucleosides. In that regard, it belongs in a more specialized journal. To argue for broader interest suitable for this journal, the authors should demonstrate greater utility in DNA/RNA.

12. Minor points:

Table 2 legend should mention reaction temperature and wavelength of LED.

A number of spelling/grammatical errors need to be corrected. For example, in paragraph 1, “minicing” and “wildly” are incorrect.

Reviewer #1

Thanks for your valuable comments and recommendations for publication after the following issues to be addressed. We have followed referee's advices and made the appropriate changes:

1) What is for me a bit difficult to believe is the reaction of GTP - in our hands the NTPs are not stable for 16 h in aqueous buffer - did the authors observe any hydrolysis of the triphosphate?

Under acidic conditions, as referee mentioned, guanosine triphosphate is unstable, but using our conditions, it will NOT undergo significant hydrolysis, and the target product can still be obtained with good yield. We believe that whether guanosine triphosphate and its product undergo significant hydrolysis depends on conditions such as acid concentration, reaction time, storage time, storage condition (e.g., inert gas protection), and so on.

Using our original conditions (Table S9, Entry 1, TFA=1.0 equiv.), hydrolysis was observed. The target product (isopropylated GTP) and hydrolysis product (isopropylated GDP) were produced in yields of 65% and 27%, respectively. Then, we screened several conditions to reduce the occurrence of triphosphate group hydrolysis. Firstly, without using TFA (Table S9, Entry 2), the yield of the target product can be increased by 5% and hydrolysis product can be decreased by 3%, compared to the original condition. Furthermore, we also attempted to use different types of buffer solutions as the aqueous phase. After replacing water with the buffer solutions (Table S9, Entry 3-5), 1% to 3% increases can be achieved, while the data discrepancies of the target product yield are not significant between them. However, when using the 10 mM NH_4HCO_3 (Table S9, Entry 5), the hydrolysis product is only 14%, which is the lowest of the three. Ulteriorly, increasing the concentration of NH_4CO_3 to 20 mM (Table S9, Entry 6), it is hard to further inhibit the occurrence of hydrolysis.

Table S9. Buffer screening

Entry	Solvent	Additive	Yield of 3 ^b	Yield of 4 ^b	pH value ^c (after reaction)
1	MeCN:H ₂ O=1:1	TFA (1 equiv.)	65%	27%	0.5
2	MeCN:H ₂ O=1:1	-	70%	24%	1.0
3	MeCN:10mM Tris-HCl (pH7.0)=1:1	-	73%	17%	1.5
4	MeCN:10mM PBS (pH7.2)=1:1	-	73%	20%	1.5
5	MeCN:10mM NH ₄ HCO ₃ =1:1	-	71% (43% ^d)	14%	1.5
6	MeCN:20mM NH ₄ HCO ₃ =1:1	-	70%	14%	1.5

^a Reaction condition: GTP (0.05 mmol, 1.0 equiv.), isopropylboronic acid (4.0 equiv.), MesAc (0.0025 mmol, 0.05 equiv.), (NH₄)₂S₂O₈ (0.1 mmol, 2.0 equiv.), catechol (0.05 mmol, 1.0 equiv.), MeCN (0.25 mL), aqueous phase (0.5 mL), irradiated by 85 W white light at r.t. for 16 h. ^b Yields were determined by HPLC. ^c pH values were determined by pH test paper. ^d Isolated yield.

Figure S7-2. Reverse-phase HPLC traces of (A) the standard sample of GTP, (B) the standard sample of C8-isopropylated GTP, (C) reaction mixture following catechol-promoted photoredox C–H alkylation of GTP with isopropyl boronic acid (specific conditions are shown in **Entry 1** in **Table S9**), (D) reaction mixture following catechol-promoted photoredox C–H alkylation of GTP with isopropyl boronic acid (specific conditions are shown in **Entry 2** in **Table S9**), (E) reaction mixture following catechol-promoted photoredox C–H alkylation of GTP with isopropyl boronic acid (specific conditions are shown in **Entry 3** in **Table S9**), (F) reaction

mixture following catechol-promoted photoredox C–H alkylation of GTP with isopropyl boronic acid (specific conditions are shown in **Entry 4** in **Table S9**), (G) reaction mixture following catechol-promoted photoredox C–H alkylation of GTP with isopropyl boronic acid (specific conditions are shown in **Entry 5** in **Table S9**), and (H) reaction mixture following catechol-promoted photoredox C–H alkylation of GTP with isopropyl boronic acid (specific conditions are shown in **Entry 6** in **Table S9**)

2). For reactions of ONs - the authors have only MS analysis and they just assume the substitution proceeded on G (rather than on other nucleobases) - they should prove it at least in 1-2 cases

Thanks for the good advice. We added two more cases that can prove the substitution proceeded on guanine. Firstly, a top-down detection method for oligonucleotides based on MALDI-FTMS was used. We reselected the shorter **ON 11** as a precursor ion, making the results more concise and conducive to analysis. The ethylated **ON 11** (precursor ion) underwent fragmentation in different forms. Its MS/MS fragmentation information is shown in **Figure S41-2**, and they inferred structural formulas are shown in **Scheme S25**. Based on the above information, it can be concluded that the site where ethylation occurs is guanine.

Furthermore, we also used a bottom-up detection method for oligonucleotides based on nuclease enzymatic hydrolysis and HPLC. The results are shown in **Figure S42-2**. Firstly, we separated the ethylated **ON 11**, and the comparison figure between the HPLC spectrums of reaction mixture and the obtained target product is shown in **Figure S42-1**. Then, the ethylated **ON 11** was hydrolyzed into nucleosides using the Nucleoside Digestion Mix (NEB #M0649), and the reaction mixture was analyzed by HPLC. The same analytical method was applied to C8-ethyl-2'-deoxyguanosine, 2'-deoxycytidine, and thymidine. Next, compare their HPLC spectrums. The result is shown in **Figure S42-2**, indicating that mixed solution of enzymatic hydrolysis reaction contains C8-ethyl-2'-deoxyguanosine, 2'-deoxycytidine, and thymidine. This can also serve as evidence that ethylation occurs at the C-8 position of guanine.

(1) Alkylation of oligonucleotides ON 11 with ethylboronic acid

Figure S41-1. Mass spectrum (MS) of ethylated ON 11, analysis by MALDI-FTMS.

Figure S41-2. MS/MS fragmentation of ethylated ON 11, analysis by MALDI-FTMS.

Scheme S25

(2) Enzyme digestion of C8-ethylated ON 11

Isolation:

Figure S42-1. Reverse-phase HPLC traces of (A) product mixture following catechol-promoted photoredox C-H alkylation of ON 11

11 with ethyl boronic acids, and (B) the isolated product of **ON 11** with ethylation.

Digestion

Figure S42-2. Reverse-phase HPLC traces of (A) product mixture following nuclease digestion (M0649S) of the isolated product of **ON 11** with ethylation, (B) the standard sample of C8-Et-deoxyguanosine, (C) the standard sample of 2'-deoxycytidine, and (D) the standard sample of thymidine.

3) The authors should better discuss the outcome of the reactions of ONs - related cross-coupling reactions on ONs strongly depend on the sequence and secondary structure - see for example:

<https://doi.org/10.1002/cbic.202100608>

<https://doi.org/10.1021/acs.bioconjchem.0c00466>

<https://doi.org/10.1093/nar/gkv185>

The authors should cite and discuss these papers - they show differences between internal and terminal position and between single- and double stranded DNA (basically cross-couplings do not proceed on dsDNA)

I highly recommend that the authors also try the C-H activation on dsDNA with G in internal and terminal position to verify if the transformation is suitable for modification of DNA.

Thanks for the good advice. We have cited these references and discussed in the text. As referee suggested,

we have also conducted the following reactions regarding the ethylation of dsDNA (**Table S20**). The results indicate that this ethylation protocol might also be able to modify guanine at various sites of dsDNA. The reactions were run at 10 °C (below its annealing temperature). For **Table S20, Entry 3**, from the results of HPLC (**Figure S39-1**), there are new peaks appearing. From the mass results, both chains were found to have ethylated products generated (**Figure S39-2**). The G on these two strands is in no special position, both in internal and terminal position, which might mean that this method can be directly used for modifying dsDNA without being affected by its pairing.

Table S20. Substrate scope of C–H ethylation of the dsDNA oligonucleotides

Entry	Oligonucleotides & Sequence	Calcd. molecular weight	Exptl. m/z (LTQ-XL)
1	ON 20 5'-d(GAT CTA TTA CGC T)-3' 3'-d(CTA GAT AAT GCG A)-5'	3952.6621 ^b	3952.9
		3980.7161 ^c	N.D.
		4010.7171 ^b	4011.2 [M+H] ⁺
		4038.7711 ^c	N.D.
		4066.8251 ^d	N.D.
2	ON 20-1 5'-d(GAT CTA TTA CGC T)-3'	3952.6621 ^b	3952.8
		3980.7161 ^c	3982.1 [M+2H] ⁺
		4010.7171 ^b	4009.0 [M-2H] ⁻
3	ON 20-2 3'-d(CTA GAT AAT GCG A)-5'	4038.7711 ^c	N.D.
		4066.8251 ^d	N.D.

^a Condition: Oligonucleotide (100 nmol), ethylboronic acid (400.0 equiv.), MesAcr (50 mol%), (NH₄)₂S₂O₈ (200.0 equiv.), TFA (50.0 equiv.), and catechol (100.0 equiv.) in MeCN (0.25 mL), H₂O (0.25 mL), irradiated by 10 W blue LED at 4 °C for 16 h. ^b Data of monosubstituted product(s). ^c Data of disubstituted product. ^d Data of trisubstituted product.

Figure S39-1. Reverse-phase HPLC traces of (A) ON 20_a prepared by solid-phase DNA synthesis, (B) ON 20-1_a prepared by solid-phase DNA synthesis, (C) ON 20-2_a prepared by solid-phase DNA synthesis, and (D) reaction mixture following catechol-

promoted photoredox C–H alkylation of **ON 20** with ethyl boronic acids.

Figure S39-2. Identification of G modification by LTQ XL analysis of the reaction system of **ON 20**. Mass spectrum (MS) of desired product after ethylation of **ON 20**.

Figure S39-3. Reverse-phase HPLC traces of (A) **ON 20-1** prepared by solid-phase DNA synthesis, and (B) reaction mixture following catechol-promoted photoredox C–H alkylation of **ON 20-1** with ethyl boronic acids.

Figure S39-4. Identification of G modification by LTQ XL analysis of the reaction system of **ON 20-1**. Mass spectrum (MS) of desired product after ethylation of **ON 20-1**.

Figure S39-5. Reverse-phase HPLC traces of (A) ON 20-2_a prepared by solid-phase DNA synthesis, and (B) reaction mixture following catechol-promoted photoredox C–H alkylation of ON 20-2 with ethyl boronic acids.

Figure S39-6. Identification of G modification by LTQ XL analysis of the reaction system of ON 20-2. Mass spectrum (MS) of desired product after ethylation of ON 20-2.

4). Cross-coupling reactions on cyclic-dinucleotides also should be cited:
<https://doi.org/10.1021/acs.jmedchem.2c01305>

Thanks for the good advice. The literature has been cited. We also used our method to modify c-di-GMP, which resulted in the target disubstituted product with an isolated yield of 53%.

Reviewer #2

Thanks for your valuable comments and recommendations for publication after the following issues to be addressed. We have followed referee's advices and made the appropriate changes:

1) While pyrimidine oligonucleotides are rather stable in TFA, this is much less the case for purine containing sequences. Usually, DNA oligonucleotides containing dA residues are depurinated and eventually hydrolyzed in high proportions upon incubation with TFA even for shorter periods of time (see e.g. 10.1080/15257770701508216; 10.1021/acs.analchem.9b03625). It is thus difficult to rationalize how ONs 12, 14, and 16 can survive 17h incubation in the presence of 50 equivalents of TFA and yield the expected product in over 55% yields. A thorough investigation on the stability and integrity of dA containing DNA oligonucleotides needs to be provided to support these findings and conclusions.

We believe that because the concentration of TFA in our conditions is particularly low (0.2%), it would not cause significant depurination of dA residues. Relevant experiments were designed to support that. **ON 13** was used as the substrate, and was placed at 0.2% TFA (50 equiv.) (**Entry 1**) and 95% TFA (**Entry 2**), respectively. We analyzed them by LC-MS, and the LC spectrum is shown in the following figure. The figure shows that the original conditions (0.2% TFA) did not lead to the hydrolysis of **ON 13** (**Figure (B)**). However, under the conditions of 95% TFA, it is evident that there are many new peaks in the LC spectrum (**Figure (C)**), which means the generation of new compounds, and molecular weight of **ON 13** is not observed.

Figure. Reverse-phase HPLC traces of (A) the standard sample of **ON 13**, (B) **Entry 1**, reaction mixture following the treatment of 50 equiv. TFA (0.2%) in MeCN/H₂O (total volume: 0.5 mL), and (C) **Entry 2**, reaction mixture following the treatment of 95% TFA in H₂O (total volume: 0.5 mL).

2). Page 7: PS bonds are known to be a redox site that can compete with G oxidation and be oxidized by a variety of radicals (see e.g. NAR 2019, 47, 11514). The authors need to provide an explanation for the absence of reactivity of the PS bonds under these reaction conditions.

We believe that the reason why the absence of reactivity of the PS bonds will happen under persulfate and light conditions is that generated free radicals first react with ethylboronic acid and are mostly consumed. Based on our inference, the following reaction was designed. The **Entry 1** is the original condition, and the **Entry 2** is that the original condition does not add ethylboronic acid. From the results, the **Entry 1** did not observe the generation of P-O related products, while the **Entry 2** generated a large amount of GpU, as well as the guanosine-3'-monophosphate.

Entry	EtB(OH) ₂ -eq	Recovery (%)	Et-Yield (%)	GpU (%)	GMP (%)
1	4	5	63	0	0
2	0	0	/	50	50

Entry1:

Entry2:

Figure. Reverse-phase HPLC traces of (A) the standard sample of GpU(S), (B) Entry 1, product mixture following catechol-promoted photoredox C-H alkylation of GpU(S) with ethyl boronic acids, (C) Entry 2, product mixture following catechol-promoted photoredox C-H alkylation of GpU(S) without ethyl boronic acids, and (D) the standard sample of GpU.

Possible transformation with or without EtB(OH)₂

3). It is often desirable to functionalize the C-8 position of purines with functional groups such as amines, thiols, imidazoles, or carboxylic acids (see e.g. NAR 2009, 37, 1638; JACS 2004, 126, 12720; JACS 2021, 143, 18960; PNAS 2010, 107, 20720). The authors should evaluate the possibility of introducing such functional groups at position C-8 of guanine.

Thanks for the good advice. We tried several R-Bpin with functional groups, and supplemented the data of R-Bpin with azide group and ester group. The results are listed as follows, as well as two not successful examples. The products with azide group and ester group can be obtained with yields of 52% and 56%, respectively. Moreover, the compound **3m** can be used as a substrate for further click reaction. We introduced the biotin alkyne with an isolated yield of 85% (LC-yield: 99%) (see Supporting Information *Scheme S23-1*).

Unsuccessful examples

4). Scheme S13-2 should probably be moved to the main manuscript since this is a rather important piece of information.

Thanks for the advice. We have followed referees' suggestion and added more data and discussion in the text, due to space limitation, we put this mechanism to the SI. If space allows, we will move the mechanism to the

main text.

5) Structure of compound 4g' should be provided in the ESI.

Regarding this, it was our mistake. In the supporting information, the data we placed is indeed **Compound 4g'**. However, 4g' was written as 4g. A correction has been made in the corresponding area now and is displayed below. At the same time, the raw ESI data of **Compound 4g'** is also placed below.

“2-amino-8-ethyl-1,9-dihydro-6H-purin-6-one (Compound 4g')

4g' was obtained following the general procedure **D**, while the reaction was performed without the TFA. After purification by column chromatography (C18 Spherical silica) using MeOH/H₂O as the eluents, **4g'** was obtained as a white solid (12.7 mg, 71%). mp >280 °C. ¹H NMR (400 MHz, DMSO-*d*₆) δ 12.27 (d, *J* = 176.7 Hz, 1H), 10.59 (s, 1H), 6.27 (d, *J* = 38.9 Hz, 2H), 2.59 (dd, *J* = 15.1, 7.8 Hz, 2H), 1.20 (t, *J* = 7.7 Hz, 3H). ¹³C NMR (176 MHz, DMSO-*d*₆) δ 156.6, 153.2, 152.6, 149.0, 115.7, 21.8, 12.2. HRMS-ESI *m/z* calcd. for C₇H₉N₅NaO [*M*+Na]⁺ 202.0699. found 202.0697.”

6). The authors need to specify in each case how much (mg and mmoles) starting material were engaged in each reaction and not only provide final yields.

We have made modifications to this suggestion in the Supporting Information. They are present below:

(1) 3.1.3. Decarboxylative borylation for the preparation of 2m” [8a, 8b] :

“A 50 mL Schlenk tube equipped with a stir bar was charged with 5-Hexynoic acid (560.2 mg, 551 μL, 1.0 equiv), N-hydroxy-phthalimide (NHPI) (815.6 mg, 1.0 equiv). Dichloromethane was added (25 mL, 0.2 M) followed by DIC (631.0 mg, 1.0 equiv), and the mixture was allowed to stir vigorously at room temperature for 45 minutes. The mixture was filtered (over Celite, SiO₂, or through a fritted funnel) and rinsed with additional CH₂Cl₂/Et₂O. The solvent was removed under reduced pressure, and purification by flash-column chromatography (eluents: DCM/ petroleum ether = 1/1) afforded the corresponding redox-active ester **2m”-1** as a white solid (983.0 mg, 76%). [8a]

A solution of the **2m”-1** (257.1 mg, 1.0 equiv), [Ir(ppy)₂dtbpy]PF₆ (9.1 mg, 1 mol%), and B2pin₂ (1.0 g, 4.0 equiv.) in DMF/MeCN/H₂O (1/1/1, 5 mL) was added into a flame-dried Schlenk tube containing a magnetic stirring bar under N₂ environment. The reaction mixture was irradiated using a 45 W white light for 14h. After completion of the reaction as monitored by TLC, the reaction mixture was diluted by H₂O and then extracted with diethyl ether (3x). The combined organic layers were dried over anhydrous MgSO₄. After the filtrate was condensed under the reduced pressure, the crude product was purified by flash-column chromatography using petroleum ether (boiling range 30-60 °C): diethyl ether (30: 1) as eluents to afford **2m”** as a colorless oil (147.5 mg, 31%) [8b].”

(2) 3.1.5. Deoxygenative borylation method for preparation of 2x” [10a, 10b] :

“The *tert*-amyl alcohol (176.3 mg, 219 μL, 1.0 equiv.), DMAP (22.4 mg, 0.1 equiv.) and Et₃N (242.4 mg, 333 μL, 1.2 equiv.) were dissolved in CH₂Cl₂ (20 mL, 0.1 M), and methyl 2-chloro-2-oxoacetate (292.8 mg, 220 μL, 1.2 equiv.) was added slowly at 0 °C. The mixture was allowed to warm to room temperature over 18 hours. Et₂O was added and the precipitate was filtered off over a plug of silica. Concentration in vacuo and subsequent purification by flash chromatography delivered the pure methyl oxalates **2x”-1** as colorless oil (319.0 mg, 91%) [10a].”

In a Schlenk tube equipped with a stir bar, **2x''-1** (69.6 mg, 1.0 equiv.), B₂cat₂ (285.4 mg, 3.0 equiv.) and fac. Ir(ppy)₃ (2.6 mg, 1.0 mol%) were dissolved in DMF (1.2 mL, 3.3 M). The solution was degassed by three freeze-pump-thaw cycles. Then, the mixture was irradiated for 15 h using a 24 W blue LED (the distance between the tube and the light source was about 1.0 cm). The setup was simultaneously cooled by a fan to keep the reaction mixture at room temperature. After completion of the reaction, Pinacol (189.0 mg, 4.0 equiv.) and Et₃N (1.9g, 2.6 mL, 47.0 equiv.) were added and stirring was continued for 2 hours. A half saturated NH₄Cl solution was added and the aqueous layer was extracted with EtOAc (3x). The combined organic layers were washed with brine and dried over MgSO₄. Concentration in vacuo and subsequent purification by flash chromatography delivered the pure **2x''** as colorless oil (143.0 mg, 90%) ^[10a].

7). A thorough discussion section needs to be included in this manuscript.

Thanks for the good suggestion, we have added the discussion part in the end of this paper.

Reviewer #3

Thanks for your valuable comments. We have followed referee's advices and made the appropriate changes:

1). In their introductory examples of late-stage functionalization of purines in DNA/RNA, the authors have missed two significant newer chemistries that deserve to be mentioned and cited. The first is the use of kethoxal and derivatives to react with guanine (PMID: 32015521) and a second is the use of SNAr chemistry to modify iodopurine (PMID: 35119264).

Thanks for good advice. We have put these two references in the Fig. 1c

2). I am not convinced that this reaction method has broad utility for chemists who modify DNA/RNA. The authors note a number of possible applications of C-8 modification in the introduction, but these may not apply to their chemistry. For example, fluorescent nucleosides with modifications at C-8 require aryl or alkenyl/alkynyl substitution to extend conjugation, and this is not possible with the new chemistry, which seems limited to alkyl substitution. This is a limitation of the new chemistry, since the earlier Suzuki chemistry allows for aryl and alkenyl substitution.

We believe that introduction of alkyl group at the C8 position of guanosine is a good supplement to the previous aryl/alkenyl/alkynyl substitution since there are rare examples of this kind of modification in the DNA/RNA. Using this new method, modified RNA/DNA oligonucleotides with varieties of of alkyl groups can be easily obtained *via* the late-stage modification approach, or solid phase synthesis with alkylated monomer. These modified oligonucleotides may have the potential application for medicinal chemistry and chemical biology: 1) to study the effects of alkyl modification on the interactions between nucleic acid with protein, such as post-selex modification of aptamer. 2) to study the effects of the alkyl modification on the activity, stability and toxicity in nucleic acid drugs, such as ASO and siRNA. 3) bioconjugation of the oligonucleotide using click chemistry for labeling and cross-linking with other biomacromolecules. Indeed we have conducted reactions on the application of modified oligonucleotide (**compound 18**) with azide group. Firstly, a group containing azide is introduced, followed by further click reactions to conjugate a biotin alkyne. The results are shown as follows.

Fig. 6 | Labeling oligonucleotide using alkylation and click reaction.

3). One additional limitation of the new chemistry for RNA/DNA is that it appears to require strict inert atmospheres with freeze/thaw degassing and Schlenk apparatus. This also lowers the chances of broader adoption.

Thanks for the suggestions. This radical reaction tolerates the water and is not compatible with the air, which need run under strict inert atmospheres without O₂. Besides the freeze/thaw degassing procedure, bubbling the solvent with N₂ to remove O₂ from the reaction system or using a Glove Box is also working. One of the coauthors, undergraduate student Ms. Yunxi Han can perform this reaction well. Definitely, we will make this reaction more practical for biologists and biochemists, such as short reaction time and easy to operate.

4). Second, C-8 modification of guanine is usually not very compatible with DNA/RNA biochemical function. Modifying the C-8 position results in altering purine geometry to the nonstandard syn conformation, which strongly hinders its ability to undergo base pairing. This also prevents it from being a good substrate for polymerase enzymes.

Thanks for the good advice. As this referee mentioned C-8 modification will result in altering purine geometry to the nonstandard syn conformation, which strongly hinders its ability to undergo base pairing, such as 8-Ph-dATP is too bulky for the polymerase to accept as a substrate. There are reports showing that 8-substituted purine derivatives bearing small groups, like amino, bromo, or methyl and the bulkier 8-[(2-imidazol-4-ylethyl)amino] are generally good substrates for DNA polymerases. Due to the hardly accessible of NTPs bearing other alkyl groups, fewer studies were reported on these functionalized oligonucleotides. After preparation of alkyl functionalized NTPs using our one-step method, we will study their compatibility with DNA/RNA polymerase and further application. Some of NTPs with small alkyl groups have the potential to be used as a substrate for polymerase. In addition, directed evolution of enzymes is currently a cutting-edge field, and that could provide a possible approach for these alkylated substrates.

In some cases, if the C-8 modified guanine is not a good substrate for polymerase enzymes, late-stage functionalization of the original oligonucleotide could be a better approach.

5). Finally, the oligonucleotides tested in Table 2 only contain one guanine base, thus guaranteeing reaction at one preferred site. However, this does not seem practically useful, as nearly all applications of DNA/RNA will require more than one guanine (usually many of them). Can the authors suggest a useful application where the chemistry could be helpful when there are several or many guanines in DNA or RNA? If they can make a strong argument for this, or demonstrate utility of this, it could add significantly to the importance and broad interest of this work.

Thanks for the good advice. We have carried out the experiment with DNA oligonucleotides (5'-dTGTCGC-3') containing two guanines as substrate. It was found that monosubstituted and disubstituted products can be obtained with yields of 33% and 67%, respectively. Next, we used RNA oligonucleotides containing two guanines and one adenine (5'-CGAUGU-3') as substrates. The results showed two monosubstituted products were generated, with a total yield of 33%, and a disubstituted product was generated, with a yield of 56%.

Entry	Oligonucleotides & Sequence	Yield [%] ^b	Calcd. Exact mass	Exptl. m/z (MALDI-TOF)
1	ON 18	33 ^c /67 ^d	1810.3654 ^c /1838.3967 ^d	[M+H] ⁺ =1811.37688 ^c /1839.41817 ^d

^a Condition: Oligonucleotide (100 nmol), ethylboronic acid (400.0 equiv.), MesAcr (50 mol%), (NH₄)₂S₂O₈ (200.0 equiv.), TFA (50.0 equiv.), and catechol (100.0 equiv.) in MeCN (0.25 mL), H₂O (0.25 mL), irradiated by 85 W white light at r.t. for 16 h. ^b Yields were determined by LC-MS and used the analytical method B. ^c Data of monosubstituted product(s). ^d Data of disubstituted product.

According to the following table, there are more than ten ASO drugs containing one G or two G in clinical or preclinical trials, moreover, our method can also tolerate the PS bond, so our method has the potential for the late-stage C8-modification of these ASO drugs, such as Tominersen for treatment of Huntington's disease (HD). In some cases, if specific G in the oligonucleotides bearing many G (more than 2), such as Nusinersen for treatment of spinal muscular atrophy (SMA), needs to be alkylated, solid-phase synthesis using alkylated monomers would be a more suitable approach.

Entry	Product Name	Chemical Structure	Number of G	Phase	Indication
1	Miravirsen	RNA, 5'-(P-thio)((2'-O,4'-C-methylene)m5C-dC-(2'-O,4'-C-methylene)A-dT-dT-(2'-O,4'-C-methylene)G-(2'-O,4'-C-methylene)m5U-dC-dA-(2'-O,4'-C-methylene)m5C-dA-(2'-O,4'-C-methylene)m5C-dT-(2'-O,4'-C-methylene)m5C-(2'-O,4'-C-methylene)m5C)-3'	1	II	Hepatitis C Virus (HCV) Infection
2	Trecovirsen	DNA, 5'-d(P-thio)(C-T-C-T-C-G-C-A-C-C-C-A-T-C-T-C-T-C-T-C-C-T-T-C-T)-3'	1	/	HIV
3	FITC-Trecovirsen sodium	DNA, 5'-d(P-thio)(C-T-C-T-C-G-C-A-C-C-C-A-T-C-T-C-T-C-T-C-C-T-T-C-T)-3', (FITC-labeled), sodium salt	1	/	HIV
4	SPC5001	5'-T-G-mC-T-A-C-A-A-A-A-C-mC-mC-A-3'	1	I	Hypercholesterolemia
5	RG-101	RNA, ((2'-O-MOE)A-sp-(2'-O-MOE)m5C-sp-(2'-O-MOE)A-sp-(2'-O-MOE)m5C-sp-(2'-O-MOE)m5C-sp-(2'-O-MOE)A-sp-(2'-O-MOE)m5U-sp-dT-sp-dG-(3'→4')-sp-[2',5'-anhydro-6'-deoxy-4'-C-(hydroxymethyl)-α-L-mannofurano]U-(3'→4')-sp-[2',5'-anhydro-6'-deoxy-4'-C-(hydroxymethyl)-α-L-mannofurano]C-sp-dA-(3'→4')-sp-[2',5'-anhydro-6'-deoxy-4'-C-(hydroxymethyl)-α-L-mannofurano]C-sp-dA-(3'→4')-sp-[2',5'-anhydro-6'-deoxy-4'-C-(hydroxymethyl)-α-L-mannofurano]C-sp-dT-(3'→4')-sp-[2',5'-anhydro-6'-deoxy-4'-C-(hydroxymethyl)-α-L-mannofurano]C-(3'→4')-sp-[2',5'-anhydro-6'-deoxy-4'-C-(hydroxymethyl)-α-L-mannofurano]C-dA), 3'-[[[(2S,4R)-1-[29-[[2-(acetylamino)-2-deoxy-β-D-galactopyranosyl]oxy]-14,14-bis[[3-[[[5-[[2-(acetylamino)-2-deoxy-β-D-galactopyranosyl]oxy]-1-oxopentyl]amino]propyl]amino]-3-oxopropoxy]methyl]-1,12,19,25-tetraoxo-16-oxa-13,20,24-triazanonacos-1-yl]-4-hydroxy-2-pyrrolidinyl]methyl hydrogen phosphate]	1	/	Chronic Hepatitis C Virus Infection
6	Tominersen	DNA, 5'-d((2'-O-MOE)m5rC-sp-(2'-O-MOE)m5rU-(2'-O-MOE)m5rC-sp-(2'-O-MOE)rA-(2'-O-MOE)rG-sp-T-sp-A-sp-A-sp-m5C-sp-A-sp-T-sp-T-sp-G-spA-sp-m5C-sp-(2'-O-MOE)rA-(2'-O-MOE)m5rC-(2'-O-MOE)m5rC-(2'-O-MOE)rA-sp-(2'-O-MOE)m5rC)-3'	2	III	Huntington's Disease (HD)
7	FITC-labeled Tominersen	FITC-labeled Tominersen (Entry 6)	2	/	Huntington's Disease (HD)
8	Ulefnersen	DNA, 5'-d((2'-O-MOE)rG-sp-(2'-O-MOE)m5rC-(2'-O-MOE)rA-(2'-O-MOE)rA-(2'-O-MOE)m5rU-G-sp-T-sp-	2	III	Amyotrophic Lateral Sclerosis (ALS)

		m5C-sp-A-sp-m5C-sp-m5C-sp-T-sp-T-sp-m5C-sp-(2'-O-MOE)rA-(2'-O-MOE)m5rU-(2'-O-MOE)rA-sp-(2'-O-MOE)m5rC-sp-(2'-O-MOE)m5rC) -3'			
9	Cobomarsen	RNA, 5'-(P-thio)((2'-O,4'-C-methylene)m5C-dA-(2'-O,4'-C-methylene)m5C-dG-dA-(2'-O,4'-C-methylene)m5U-(2'-O,4'-C-methylene)m5U-dA-(2'-O,4'-C-methylene)G-dC-(2'-O,4'-C-methylene)A-(2'-O,4'-C-methylene)m5U-(2'-O,4'-C-methylene)m5U-(2'-O,4'-C-methylene)A)-3'	2	II	Cutaneous T-Cell Lymphoma/My cosis Fungoides
10	Zilganersen	DNA, 5'-d((2'-O-MOE)m5rC-sp-(2'-O-MOE)rA-(2'-O-MOE)rG-(2'-O-MOE)m5rU-(2'-O-MOE)rA-(2'-O-MOE)m5rU-T-sp-A-sp-m5C-sp-m5C-sp-T-sp-m5C-sp-T-sp-A-sp-m5C-sp-T-sp-(2'-O-MOE)rA-(2'-O-MOE)rG-sp-(2'-O-MOE)m5rU-sp-(2'-O-MOE)m5rC)-3'	2	III	Alexander Disease
11	Bezeparsen	DNA, d(P-thio)[(2',5'-anhydro-6'-deoxy-4'-C-(hydroxymethyl)- α -L-mannofurano]rA-(3'→4')-[2',5'-anhydro-6'-deoxy-4'-C-(hydroxymethyl)- α -L-mannofurano]rA-(3'→4')-[2',5'-anhydro-6'-deoxy-4'-C-(hydroxymethyl)- α -L-mannofurano]m5rU-A-A-T-m5C-T-m5C-A-T-G-T-(3'→4')-[2',5'-anhydro-6'-deoxy-4'-C-(hydroxymethyl)- α -L-mannofurano]m5rC-(3'→4')-[2',5'-anhydro-6'-deoxy-4'-C-(hydroxymethyl)- α -L-mannofurano]rA-(3'→4')-[2',5'-anhydro-6'-deoxy-4'-C-(hydroxymethyl)- α -L-mannofurano]rG), 5'-[26-[[2-(acetylamino)-2-deoxy- β -D-galactopyranosyl]oxy]-14,14-bis[[3-[[6-[[2-(acetylamino)-2-deoxy- β -D-galactopyranosyl]oxy]hexyl]amino]-3-oxopropoxy]methyl]-8,12,19-trioxo-16-oxa-7,13,20-triazahexacos-1-yl hydrogen phosphate]	2	/	PCSK9 synthesis inhibitor.
12	RGLS4326	5'-AsGsCmAfCfUfUmUsGs-3'	2	I	Polycystic Kidney Disease, Autosomal Dominant

Post-modification

Oligonucleotides with *less than two guanines*

Solid-phase synthesis

Oligonucleotides with *more than two guanines*

Examples of proposals for synthesizing the C8-alkyl-guanosine oligos.

6). In addition, post-synthesis modification of an oligo by the authors' approach will require HPLC purification to separate the product from the significant amount of unreacted starting material or other byproducts. Thus, it seems more likely that their chemistry would have more value by simply modifying the nucleoside monomer and incorporating it into synthetic oligonucleotides by solid phase synthesis. By that method, nearly 100% modification would be achieved.

Thanks for the advice. Indeed, our methods can provide different nucleoside monomers for solid-phase synthesis. If specific alkylation of G oligonucleotides is wanted, or specific G in the oligonucleotides bearing

many G (more than 3) needs to be alkylated, this method may be a good approach *via* our nucleoside chemistry. However, in some cases, late-stage functionalization may have advantages. For example, it is hard to introduce bulky or sensitive groups in the C8 position using solid-phase synthesis. Moreover, if it wants to introduce various of different functional groups for C-8 guanine on oligonucleotides for screening purposes, late-stage modification commercially available or natural oligonucleotides could easily obtain a library of oligonucleotides. In contrast, solid-phase synthesis approach needs to first introduce different functional groups, and then undergo a series of reactions to produce corresponding phosphoramidite (such products can be used as substrates for *de novo* synthesis). For example, ten different groups need to be introduced, then ten *de novo* synthesis steps (at least 60 steps, total) of oligonucleotides are required, which wastes time and resources. Using our method, although HPLC separation is required, only ten reactions are needed, which is more efficient, especially for the initial screening of the different modified oligos.

Different approaches for the synthesis of the C8-alkyl-guanosine oligos.

7). Finally, the authors only documented the addition of an ethyl group at C8 of G in RNA/DNA. If the utility in RNA/DNA is to be supported, then the authors should demonstrate reaction with a broader range of modifications, including ones that biochemists or, for example, siRNA medicinal chemists, might find useful.

This is a very good suggestion. Since there are rare examples of C8-alkylated oligonucleotides reported, these various C8-alkylated oligonucleotides may have applications in the area of nucleic acid drugs, such as ASO, siRNA and aptamer. We have conducted relevant experiments, and their data is as follows (Table 2):

Entry	Oligonucleotides & Sequence	Yield [%] ^b	Substituent group	Calcd. Exact mass	Exptl. m/z (MALDI-TOF)
13 ^e	ON 5 5'-AUUGCC-3'	59	n Pr-B(OH) ₂	1876.3243	[M+H] ⁺ = 1877.31866
14 ^f	ON 5 5'-AUUGCC-3'	31	N ₃ (CH ₂) ₃ -Bpin	1917.3257	[M+H] ⁺ = 1918.30266
15 ^g	ON 5 5'-AUUGCC-3'	57	Cyclobutyl-B(OH) ₂	1888.3243	[M+H] ⁺ = 1889.30149
16 ^h	ON 5 5'-AUUGCC-3'	63 ⁱ	Cyclopentyl-B(OH) ₂	1902.3400	[M+H] ⁺ = 1903.33673

^a General condition: Oligonucleotide (100 nmol), ethylboronic acid (400.0 equiv.), MesAcr (50 mol%), (NH₄)₂S₂O₈ (200.0 equiv.),

TFA (50.0 equiv.), and catechol (100.0 equiv.) in MeCN (0.25 mL), H₂O (0.25 mL), irradiated by 85 W white light at r.t. for 16 h. ^b Yields were determined by LC-MS and used the analytical method B. ^c Yields were determined by LC-MS and used the analytical method A. ^d The mass data was collected by using HRMS. ^e *N*-butylboronic acid (400.0 equiv.) was used, and reaction time was extended to 24 hours. ^f alkylpinacolyl boronate esters (400.0 equiv.) was used, and methylboronic acid (400.0 equiv.) was additionally added. ^g Cyclobutylboronic acid (200.0 equiv.) was used, equivalent of catechol has been reduced to 200, and irradiated by 10 W white LED at 10 °C. ^h Cyclopentylboronic acid (400.0 equiv.) was used, and irradiated by 10 W white LED at 10 °C. ⁱ Yields were determined by LC-MS and used the analytical method C.

8). In summary, I believe that this chemistry is a useful addition for making C-8 alkyl-modified guanine nucleosides. In that regard, it belongs in a more specialized journal. To argue for broader interest suitable for this journal, the authors should demonstrate greater utility in DNA/RNA.

Thanks for the comment. Indeed, our methods is practical methods for C-8 alkyl-modified guanine nucleosides, especially for the medicinal chemists to quickly set up nucleoside compound libraries for drug discovery. Besides C-8 alkyl-modified guanine nucleosides, our method can directly modify the cyclic dinucleotides (CDNs) (such as c-di-GMP in Fig. 4), and has potential to quickly provide a CDNs library for screening; Our protocol can also provide C8 alkyl functionalized NTPs for study their compatibility with DNA/RNA polymerase and further application. Moreover, C-8 alkyl-modified guanine DNA/RNA oligonucleotides could be easily obtained using our simpler method of direct post modification of oligonucleotides, or solid-phase synthesis with using alkylated monomer. These modified oligonucleotides may have the potential application for medicinal chemistry and chemical biology : 1) to study the effects of alkyl modification on the interactions beteewn nucleic acid with protein, such as post-selex modification of aptamer. 2) to study the effects of the alkyl modification on the activity, stabiliy and toxicity in nucleic acid drugs, such as ASO and siRNA. 3) bioconjugation of the oligonucleotide using click chemistry for labeling and cross-linking with other biomacromolecules.

9). Minor points:

Table 2 legend should mention reaction temperature and wavelength of LED.

A number of spelling/grammatical errors need to be corrected. For example, in paragraph 1, “minicing” and “wildly” are incorrect

Thanks for your suggestions. In response to these, we have made modifications, and the modified parts are shown below:

Table 2:

“^a General condition: Oligonucleotide (100 nmol), ethylboronic acid (400.0 equiv.), MesAcr (50 mol%), (NH₄)₂S₂O₈ (200.0 equiv.), TFA (50.0 equiv.), and catechol (100.0 equiv.) in MeCN (0.25 mL), H₂O (0.25 mL), irradiated by 85 W white light at r.t. for 16 h. ^b Yields were determined by LC-MS and used the analytical method B. ^c Yields were determined by LC-MS and used the analytical method A. ^d The mass data was collected by using HRMS. ^e *N*-butylboronic acid (400.0 equiv.) was used, and reaction time was extended to 24 hours. ^f alkylpinacolyl boronate esters (400.0 equiv.) was used, and methylboronic acid (400.0 equiv.) was additionally added. ^g Cyclobutylboronic acid (200.0 equiv.) was used, equivalent of catechol has been reduced to 200, and irradiated by 10 W white LED at 10°C. ^h Cyclopentylboronic acid (400.0 equiv.) was used, and irradiated by 10 W white LED at 10°C. ⁱ Yields were determined by LC-MS and used the analytical method C.”

The specific wavelength information of various lamps is written in general information of supporting information: “The lights and their wavelength: 45 W and 85 W white light (~ 437.2-616.2 nm) (high power energy saving lamps), 10 W white LED (450-465 nm), 10 W, 24 W, and 35 W blue LED (450-455 nm).”

Sincerely yours,
Gang Chen

REVIEWERS' COMMENTS

Reviewer #1 (Remarks to the Author):

In the revised version, the authors have addressed all the critical points and suggestions raised by myself and by the other reviewers. In particular, they studied the stability of GTP under the reaction conditions and optimized the protocol to minimize the hydrolysis. They also confirmed that guanine was alkylated on the ON. They also performed reactions of dsDNA and found that it is still reactive and they performed the reaction on CDN.

I am happy with all those added experiments and results which greatly improve the manuscript. I have only a few last small suggestions that the authors might consider in the last revision:

1. I am a bit surprised that they did not observe any difference in reactivity of ssON and dsDNA when obviously the access to the 8-C-H of G is very different. Are they sure that the dsDNA (I would not call it ON 20 - since it is dsDNA composed of two ONs) is stable as duplex under the reaction conditions ? Can the authors calculate conversion of the reactions and add the numbers to the Table 5 ?

2. in the Discussion on page 12 the authors correctly claim that the method could be applied for bioconjugations and cross-linking with other biomacromolecules - perhaps some references to recent examples could be added here (there is also a review on this topic:
<https://doi.org/10.1016/j.cbpa.2019.07.007>)

Congratulations to the authors to this excellent work!

Reviewer #2 (Remarks to the Author):

This contribution was evaluated by three independent reviewers (I was reviewer #2), who raised a number of important issues. The authors have now comprehensively addressed all the points that were raised and certainly markedly improved the quality of the manuscript. Importantly, the authors have added additional experiments including a thorough analysis of the reaction products stemming from alkylation of oligonucleotides, stability assays of RNA oligonucleotides under the acidic conditions that were employed, and additional alkylation reactions to introduce a larger variety of functional groups. Overall, this article is now vastly improved and publication in Nature Communications is recommended pending some revisions:

1) The stability of nucleotides and oligonucleotides under the acidic experimental conditions of the alkylation method were raised by both reviewers #1 and 2. The authors have now shown that oligonucleotides but they could not identify conditions that would prevent hydrolysis of the phosphodiester linkages of GTP. This is a drawback of this method since the separation of di- from triphosphates can be a challenging undertaking and this should be explicitly be stated in the manuscript.

2) Comments #4 and 8 by reviewer #3: the authors mention in their answer that "... the bulkier 8-[(2-imidazol-4-ylethyl)amino] are generally good substrates for DNA polymerases". While dATPs (and not

dGTPs) equipped with this structural motif are indeed tolerated by polymerases, they are far from being good substrates (see e.g. *Eur. J. Org. Chem.* 2008: 4915-4923). In addition, in order to demonstrate the usefulness of their approach, the authors should evaluate the substrate tolerance of the 8-modified nucleotides by polymerases.

Reviewer #3 (Remarks to the Author):

This is an extensive study, and the authors have responded to reviewers with additional experiments and with changes to the writing, improving the manuscript significantly. I am convinced that this reaction method can indeed be useful, especially with mononucleosides and mononucleotides, which justifies publication even if the impact in oligonucleotide modification may be low due to moderate yields.

One experiment needs modification in the writing: The authors attempted modification of "double-stranded DNA" oligonucleotides (Table S20, Fig. S39) under conditions of 50% acetonitrile in water with only micromolar salt levels. It is almost certain that these DNAs are not double-stranded under these conditions, but are rather reacting as separate single strands. Unless the authors can show evidence for double-stranded structure, cautionary statements should be added to the main text and the SI file stating that the cosolvent and lack of salts may not support double-stranded hybridization, and it is possible that the strands are reacting in single-stranded form.

Reviewer #1 (Remarks to the Author):

In the revised version, the authors have addressed all the critical points and suggestions raised by myself and by the other reviewers. In particular, they studied the stability of GTP under the reaction conditions and optimized the protocol to minimize the hydrolysis. They also confirmed that guanine was alkylated on the ON. They also performed reactions of dsDNA and found that it is still reactive and they performed the reaction on CDN.

I am happy with all those added experiments and results which greatly improve the manuscript. I have only a few last small suggestions that the authors might consider in the last revision:

1. I am a bit surprised that they did not observe any difference in reactivity of ssON and dsDNA when obviously the access to the 8-C-H of G is very different. Are they sure that the dsDNA (I would not call it ON 20 - since it is dsDNA composed of two ONs) is stable as duplex under the reaction conditions? Can the authors calculate conversion of the reactions and add the numbers to the Table 5?

We cannot definitively confirm whether dsDNA is stable as duplex under the reaction conditions. Although, we have made some efforts to ensure its stability and without compromising the reaction outcome. Firstly, the reaction temperature (4°C) is significantly lower than its annealing temperature (44°C). Secondly, DEPC water was utilized as the aqueous phase. However, since the cosolvent (MeCN) and lack of salts (e.g. Mg^{2+}), the dsDNA may not be duplex during the reaction. Hence, we put a cautionary statement in the manuscript and Supporting Information, mentions that “the cosolvent and lacking of salts may not support double-stranded hybridization, and it is possible that the strands are reacting in single-stranded form.”.

“From our preliminary results (Table 5, see Figure S39-2 for more detail), both chains were found to have ethylated products generated. The G on these two strands is in no special position, both in internal and terminal position, which might mean that this method can be directly used for modifying dsDNA without being affected by its pairing. However, the cosolvent and lacking of salts may not support double-stranded hybridization, and it is possible that the strands are reacting in single-stranded form.”

Table 5. Substrate scope of C–H ethylation of the dsDNA oligonucleotides ^a

Entry	Oligonucleotides & Sequence	Calcd. molecular weight	Exptl. m/z (LTQ-XL)
-------	-----------------------------	-------------------------	---------------------

		3952.6621 ^b	3952.9
	ON 20	3980.7161 ^c	N.D.
1	5'-d(GAT CTA TTA CGC T)-3'	4010.7171 ^b	4011.2 [M+H] ⁺ ^e
	3'-d(CTA GAT AAT GCG A)-5'	4038.7711 ^c	N.D.
		4066.8251 ^d	N.D.
2	ON 20-1	3952.6621 ^b	3952.8
	5'-d(GAT CTA TTA CGC T)-3'	3980.7161 ^c	3982.1 [M+2H] ⁺
		4010.7171 ^b	4009.0 [M-2H] ⁻
3	ON 20-2	4038.7711 ^c	N.D.
	3'-d(CTA GAT AAT GCG A)-5'	4066.8251 ^d	N.D.

^a Condition: Oligonucleotide (100 nmol), ethylboronic acid (400.0 equiv.), MesAcr (50 mol%), (NH₄)₂S₂O₈ (200.0 equiv.), TFA (50.0 equiv.), and catechol (100.0 equiv.) in MeCN (0.25 mL), H₂O (0.25 mL), irradiated by 10 W blue LED at 4 °C for 16 h. ^b Data of monosubstituted product(s). ^c Data of disubstituted product. ^d Data of trisubstituted product. **LTQ-XL**, Linear Ion Trap Mass Spectrometer. ^e The cosolvent and lack of salts may not support double-stranded hybridization, and it is possible that the strands are reacting in single-stranded form.

Table S20. Substrate scope of C–H ethylation of the dsDNA oligonucleotides ^a

Entry	Oligonucleotides & Sequence	Calcd. molecular weight	Exptl. m/z (LTQ-XL)
		3952.6621 ^b	3952.9
	ON 20	3980.7161 ^c	N.D.
1	5'-d(GAT CTA TTA CGC T)-3'	4010.7171 ^b	4011.2 [M+H] ⁺ ^e
	3'-d(CTA GAT AAT GCG A)-5'	4038.7711 ^c	N.D.
		4066.8251 ^d	N.D.
2	ON 20-1	3952.6621 ^b	3952.8
	5'-d(GAT CTA TTA CGC T)-3'	3980.7161 ^c	3982.1 [M+2H] ⁺
		4010.7171 ^b	4009.0 [M-2H] ⁻
3	ON 20-2	4038.7711 ^c	N.D.
	3'-d(CTA GAT AAT GCG A)-5'	4066.8251 ^d	N.D.

^a Condition: Oligonucleotide (100 nmol), ethylboronic acid (400.0 equiv.), MesAcr (50 mol%), (NH₄)₂S₂O₈ (200.0 equiv.), TFA (50.0 equiv.), and catechol (100.0 equiv.) in MeCN (0.25 mL), H₂O (0.25 mL), irradiated by 10 W blue LED at 4 °C for 16 h. ^b Data of monosubstituted product(s). ^c Data of disubstituted product. ^d Data of trisubstituted product. **LTQ-XL**, Linear Ion Trap Mass Spectrometer. ^e The cosolvent and lack of salts may not support double-stranded hybridization, and it is possible that the strands are reacting in single-stranded form.

Regarding the conversion rate, we have previously employed LC-MS for testing.

Initially, mass data were utilized to identify the compounds corresponding to which peak. Subsequently, HPLC was used to determine the ratio of peak areas at 260 nm, representing the yield of products and the recovery of raw materials. However, due to the complexity of the substrate (**ON 20**), it is challenging to ascertain the peaks corresponding to each product and the remaining raw materials. Perhaps, in the future, we need to explore a more convenient and suitable analytical method for detection."

2. in the Discussion on page 12 the authors correctly claim that the method could be applied for bioconjugations and cross-linking with other biomacromolecules - perhaps some references to recent examples could be added here (there is also a review on this topic: <https://doi.org/10.1016/j.cbpa.2019.07.007>)

Thank you for your valuable suggestions. We have cited this review in the text as ref.63.

Reviewer #2 (Remarks to the Author):

This contribution was evaluated by three independent reviewers (I was reviewer #2), who raised a number of important issues. The authors have now comprehensively addressed all the points that were raised and certainly markedly improved the quality of the manuscript. Importantly, the authors have added additional experiments including a thorough analysis of the reaction products stemming from alkylation of oligonucleotides, stability assays of RNA oligonucleotides under the acidic conditions that were employed, and additional alkylation reactions to introduce a larger variety of functional groups. Overall, this article is now vastly improved and publication in Nature Communications is recommended pending some revisions:

- 1) The stability of nucleotides and oligonucleotides under the acidic experimental conditions of the alkylation method were raised by both reviewers #1 and 2. The authors have now shown that oligonucleotides but they could not identify conditions that would prevent hydrolysis of the phosphodiester linkages of GTP. This is a drawback of this method since the separation of di- from triphosphates can be a challenging undertaking and this should be explicitly be stated in the manuscript.

Thanks for your suggestion, we have explicitly stated that in the manuscript:

“As seen in **Fig. 4b**, guanosine monophosphate (GMP) provided the desired product **8j** in good yield (LC-MS yield 91%, isolated yield 52%). However, guanosine diphosphate (GDP), and GTP afforded the corresponding C8-alkylated products (**8k** and **8l**) with low yield, along with phosphodiester bond hydrolyzed by-product under the standard condition due to the acidic reaction condition. Since 5'-polyphosphorylated nucleosides is not stable in the acidic system, reaction condition without TFA may improve the yield. However, no better results were obtained in the absence of TFA, as this system after reaction is still very acidic (the pH value was 1.97, see **Table S10**). Finally, this reaction proceeded well with moderate to good yields using the 10mM NH₄HCO₃ as solvent (pH value was 2.72 after reaction, see **Table S10**). In this condition, regrettably, we did not find a satisfactory method to prevent hydrolysis of **8l**. It is worth noting that this method requires only one-step to obtain GTP analogues with C8 substitution which efficiently inhibits FtsZ polymerization, in comparison, the previous method required four steps¹⁷.”

- 2) Comments #4 and 8 by reviewer #3: the authors mention in their answer that “... the bulkier 8-[(2-imidazol-4-ylethyl)amino] are generally good substrates for DNA polymerases”. While dATPs (and not dGTPs) equipped with this structural motif are indeed tolerated by polymerases, they are far from being good substrates (see e.g. Eur. J. Org. Chem. 2008: 4915-4923). In addition, in order to demonstrate the usefulness of their approach, the authors should evaluate the substrate tolerance of the 8-modified nucleotides by polymerases.

Thank you for your advice. This is a topic that our group will explore in future studies.

Reviewer #3 (Remarks to the Author):

This is an extensive study, and the authors have responded to reviewers with additional experiments and with changes to the writing, improving the manuscript significantly. I am convinced that this reaction method can indeed be useful, especially with mononucleosides and mononucleotides, which justifies publication even if the impact in oligonucleotide modification may be low due to moderate yields.

One experiment needs modification in the writing: The authors attempted modification of “double-stranded DNA” oligonucleotides (Table S20, Fig. S39) under conditions of 50% acetonitrile in water with only micromolar salt levels. It is almost certain that these DNAs are **not** double-stranded under these conditions, but are rather reacting as separate single strands. Unless the authors can show evidence for double-stranded structure, cautionary statements should be added to the main text and the SI file stating that the cosolvent and lack of salts may not support double-stranded hybridization, and it is possible that the strands are reacting in single-stranded form.

Thank you for your suggestion. We have incorporated this cautionary statement into the manuscript and Supporting Information.

“From our preliminary results (Table 5, see Figure S39-2 for more detail), both chains were found to have ethylated products generated. The G on these two strands is in no special position, both in internal and terminal position, which might mean that this method can be directly used for modifying dsDNA without being affected by its pairing. However, the cosolvent and lacking of salts may not support double-stranded hybridization, and it is possible that the strands are reacting in single-stranded form.”

Table 5. Substrate scope of C–H ethylation of the dsDNA oligonucleotides ^a

Entry	Oligonucleotides & Sequence	Calcd. molecular weight	Exptl. m/z (LTQ-XL)
1	5'-d(GAT CTA TTA CGC T)-3' 3'-d(CTA GAT AAT GCG A)-5'	3952.6621 ^b	3952.9
		ON 20	N.D.
		3980.7161 ^c	N.D.
		4010.7171 ^b	4011.2 [M+H] ^{+ e}
2	5'-d(GAT CTA TTA CGC T)-3'	4038.7711 ^c	N.D.
		4066.8251 ^d	N.D.
		ON 20-1	3952.6621 ^b
3	3'-d(CTA GAT AAT GCG A)-5'	3980.7161 ^c	3982.1 [M+2H] ⁺
		4010.7171 ^b	4009.0 [M-2H] ⁻
		ON 20-2	N.D.
		4038.7711 ^c	N.D.
		4066.8251 ^d	N.D.

^a Condition: Oligonucleotide (100 nmol), ethylboronic acid (400.0 equiv.), MesAcr (50 mol%), (NH₄)₂S₂O₈ (200.0 equiv.), TFA (50.0 equiv.), and catechol (100.0 equiv.) in MeCN (0.25 mL), H₂O (0.25 mL), irradiated by 10 W blue LED at 4 °C for 16 h. ^b Data of monosubstituted product(s). ^c Data of disubstituted product. ^d Data of trisubstituted product. **LTQ-XL**, Linear Ion Trap Mass Spectrometer. ^e The cosolvent and lack of salts may not support double-stranded hybridization, and it is possible that the strands are reacting in single-stranded form.

Table S20. Substrate scope of C–H ethylation of the dsDNA oligonucleotides ^a

Entry	Oligonucleotides & Sequence	Calcd. molecular weight	Exptl. m/z (LTQ-XL)
1	ON 20 5'-d(GAT CTA TTA CGC T)-3' 3'-d(CTA GAT AAT GCG A)-5'	3952.6621 ^b	3952.9
		3980.7161 ^c	N.D.
		4010.7171 ^b	4011.2 [M+H] ⁺ ^e
		4038.7711 ^c	N.D.
2	ON 20-1 5'-d(GAT CTA TTA CGC T)-3'	4066.8251 ^d	N.D.
		3952.6621 ^b	3952.8
		3980.7161 ^c	3982.1 [M+2H] ⁺
3	ON 20-2 3'-d(CTA GAT AAT GCG A)-5'	4010.7171 ^b	4009.0 [M-2H] ⁻
		4038.7711 ^c	N.D.
		4066.8251 ^d	N.D.

^a Condition: Oligonucleotide (100 nmol), ethylboronic acid (400.0 equiv.), MesAcr (50 mol%), (NH₄)₂S₂O₈ (200.0 equiv.), TFA (50.0 equiv.), and catechol (100.0 equiv.) in MeCN (0.25 mL), H₂O (0.25 mL), irradiated by 10 W blue LED at 4 °C for 16 h. ^b Data of monosubstituted product(s). ^c Data of disubstituted product. ^d Data of trisubstituted product. **LTQ-XL**, Linear Ion Trap Mass Spectrometer. ^e The cosolvent and lack of salts may not support double-stranded hybridization, and it is possible that the strands are reacting in single-stranded form.

Sincerely yours,
Gang Chen